# Pairwise is Not Enough: Hypergraph Neural Networks for Multi-Agent Pathfinding

**Rishabh Jain**[1], **Keisuke Okumura**[1,2], **Michael Amir**[1], **Pietro Liò**[1], **Amanda Prorok**[1]
[1]University of Cambridge, UK
[2]National Institute of Advanced Industrial Science and Technology (AIST), Japan
`{rj412,ko393,ma2151,pl219,asp45}@cst.cam.ac.uk`

## Abstract

Multi-Agent Path Finding (MAPF) is a representative multi-agent coordination problem, where multiple agents are required to navigate to their respective goals without collisions. Solving MAPF optimally is known to be NP-hard, leading to the adoption of learning-based approaches to alleviate the online computational burden. Prevailing approaches, such as Graph Neural Networks (GNNs), are typically constrained to *pairwise* message passing between agents. However, this limitation leads to suboptimal behaviours and critical issues, such as attention dilution, particularly in dense environments where group (i.e. beyond just two agents) coordination is most critical. Despite the importance of such higher-order interactions, existing approaches have not been able to fully explore them. To address this representational bottleneck, we introduce `HMAGAT` (Hypergraph Multi-Agent Attention Network), a novel architecture that leverages attentional mechanisms over directed hypergraphs to explicitly capture group dynamics. Empirically, `HMAGAT` establishes a new state-of-the-art among learning-based MAPF solvers: e.g., despite having just 1M parameters and being trained on $100\times$ less data, it outperforms the current SoTA 85M parameter model. Through detailed analysis of `HMAGAT`'s attention values, we demonstrate how hypergraph representations mitigate the attention dilution inherent in GNNs and capture complex interactions where pairwise methods fail. Our results illustrate that appropriate inductive biases are often more critical than the training data size or sheer parameter count for multi-agent problems.

## 1 Introduction

Multi-agent coordination is a fundamental topic in multi-agent systems, where multiple agents work together to achieve a common goal. Applications include warehouse automation (Wurman et al., 2008), autonomous driving (Shalev-Shwartz et al., 2016), traffic management (Adler & Blue, 2002) and manufacturing process control (Leng et al., 2023). These settings involve complex interactions and dependencies among agents. In the literature, such interactions are typically modelled in a pairwise manner, using graph neural networks (GNNs) or transformers (Bernárdez et al., 2023; Liu et al., 2024; Li et al., 2024b). However, pairwise interactions are insufficient to fully capture highly-coupled multi-agent dynamics, where interactions may involve multiple agents simultaneously.

This insufficiency has led to the exploration of higher-order representational structures, such as *hypergraphs*, which can naturally model group interactions. Recent works have shown that hypergraphs can effectively model small-scale multi-agent problems (Zhu et al., 2024; Zhao et al., 2025) and simple group interactions, like social groups in trajectory prediction tasks (Lin et al., 2024; Xu et al., 2022). However, the question remains, "Can hypergraphs scale beyond simple group settings to capture the dynamics of complex, highly-coupled multi-agent tasks?"

In this work, we provide a positive answer to this question by focusing on one such problem of *multi-agent pathfinding (MAPF)*, where the goal is to compute collision-free paths that efficiently guide a team of agents to their respective destinations. Optimally solving MAPF is known to be

---

Our code is available at `https://github.com/proroklab/hmagat`.

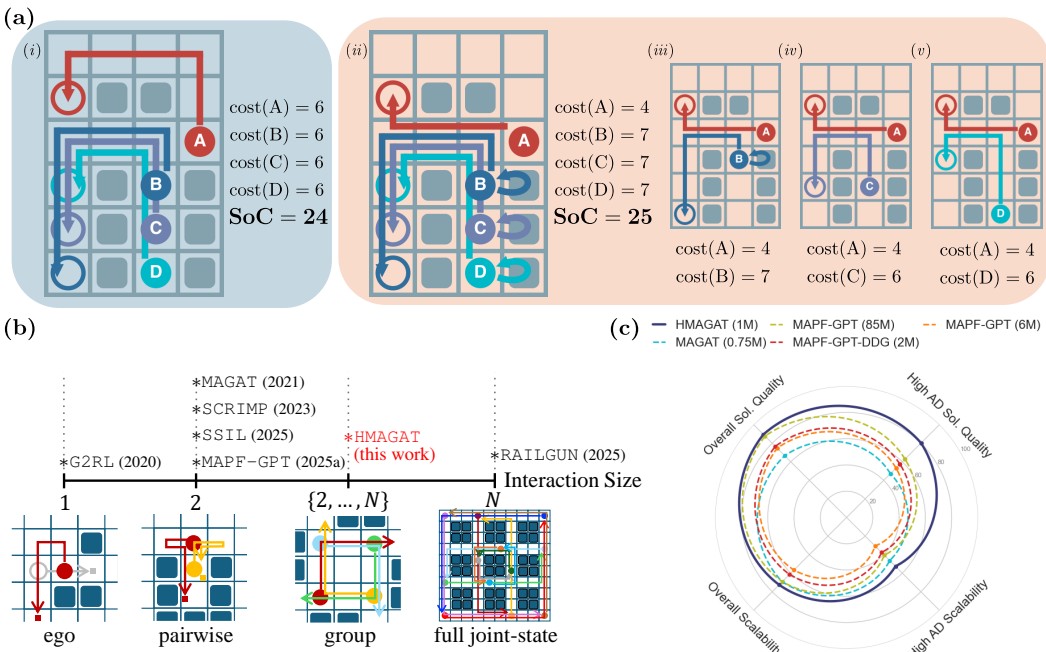

Figure 1: **(a) MAPF requires group interactions.** (*i*) The SoC-optimal (sum-of-costs) group interaction-based solution. (*iii-v*) Pairwise interactions with agent A, showing the optimal paths when agent pairs are isolated. (*ii*) Combination of the pairwise solutions. This solution is sub-optimal compared to the group interaction-based solution. **(b) Group interaction modelling has been previously unexplored.** Positioning of MAPF solvers with respect to their interaction modelling. **(c) HMAGAT achieves state-of-the-art performance.** Radar plot comparing learnt MAPF solvers, showing average solution quality and scalability (both the higher, the better) across different small maps[1] and agent densities, and a focused view on the highest agent density (High AD) scenarios. Details of the metrics are in Appendix B.

NP-hard (Surynek, 2010), even in sparse grids (Geft & Halperin, 2022), which highlights the strong entanglement among agents. This makes MAPF a suitable testbed for studying highly-coupled multi-agent problems, especially in obstacle-dense maps with large numbers of agents.

As with the other multi-agent problems, previous works on MAPF have focused on pairwise interaction-modelling, popularly using GNNs (Li et al., 2020; 2021; Veerapaneni et al., 2025) or transformers (Wang et al., 2023; Andreychuk et al., 2025a). However, MAPF is inherently a group problem, i.e., optimality and completeness can, generally, only be achieved when modelling the full joint state space of all agents (not just two), as shown in Figure 1a. We argue that the use of hypergraphs provides strong inductive biases for capturing group interactions, thereby enabling the development of more capable MAPF approaches.

To address this gap, *(i)* we propose HMAGAT, a novel imitation learning framework for MAPF, leveraging a hypergraph attention network to better model the higher-order group interactions. *(ii)* We propose hypergraph generation strategies that dynamically construct directed hypergraphs and *(iii)* empirically demonstrate the prowess of our approach over existing solvers, as shown in Fig. 1c, beating the previous state-of-the-art learning model while using only ~1.2% parameters and ~1% of the training data. *(iv)* Through experiments, we perform a deeper analysis on why pairwise attention fails while modelling group interactions, and how hypergraphs overcome these issues.

---

[1]Large maps not included in the radar plot, as MAPF-GPT fails to scale to them, resulting in skewed results.

## 2 PRELIMINARIES

### 2.1 HYPERGRAPHS AND MULTI-AGENT LEARNING

Hypergraphs are a higher-order generalisation of graphs, where edges, called *hyperedges*, can connect any number of nodes. Formally, a *directed hypergraph* is defined as $\mathcal{H} = (\mathcal{V}, \mathcal{E})$, where $\mathcal{V}$ is the set of nodes and $\mathcal{E}$ is the set of directed hyperedges. Each hyperedge $e \in \mathcal{E}$ is an ordered pair $(T(e), H(e))$, where $T(e) \subseteq \mathcal{V}$ is the tail and $H(e) \subseteq \mathcal{V}$ is the head of the hyperedge. A set of hyperedges incident to node $i \in \mathcal{V}$ is denoted as $\Gamma(i) = \{e \in \mathcal{E} \mid i \in T(e) \cup H(e)\}$.

The hypergraph counterpart of graph neural networks (GNNs) is referred to as *hypergraph neural networks (HGNNs)*. HGNNs have been increasingly adopted in multi-agent learning tasks due to their ability to capture higher-order interactions. E.g., HGNNs for formation control and robotic warehouses (Zhao et al., 2025), multi-agent continuous trajectory prediction (Lin et al., 2024; Xu et al., 2022) and social robot navigation (Li et al., 2024a; Wang et al., 2024). While these studies focus on problems with often modest team sizes ($\approx 10^1$ agents), where tightly coupled coordination among agents is less critical, our work explores the potential of HGNNs in a more challenging coupled setting containing large number of agents, namely, MAPF.

### 2.2 MULTI-AGENT PATHFINDING (MAPF)

We consider a widely used MAPF formulation (Stern et al., 2019), defined by a team of agents $A = \{1, 2, \ldots, n\}$, a four-connected grid graph $G = (V, E)$, and a distinct start $s_i \in V$ and goal $g_i \in V$ for each agent $i \in A$. At each timestep, each agent either stays in place or moves to an adjacent vertex. Vertex and edge collisions are prohibited: agents cannot occupy the same vertex simultaneously or swap their occupied vertices. Then, for each agent $i \in A$, we aim to assign a collision-free path $\pi_i = (v^0 = s_i, v^1, \ldots, v^T = g_i)$. The solution quality is assessed by *sum-of-costs* (SoC), which sums the travel time of each agent until it stops at the target location. Herein, a *configuration* $\mathcal{Q} \in V^n$ refers to the locations for all agents. The dist function returns the shortest path length between two vertices on $G$, while $\text{neigh}(v)$ represents adjacent vertices for $v \in V$.

MAPF is an appealing benchmark to assess the capabilities of multi-agent learning techniques, tailored to industrial setups such as warehouse automation (Agaskar et al., 2025). Several directions exist on how to leverage ML for MAPF (Alkazzi & Okumura, 2024), with a major thrust being the governance of agent-wise neural policies that interpret their surrounding information and output actions, ideally leading to coordination beyond mere greedy behaviour. Such policies are typically constructed either via reinforcement learning (RL) (Sartoretti et al., 2019) or imitation learning (IL) (Li et al., 2021). This work focuses on IL setups, motivated by recent findings that *(i)* collecting demonstration data is inexpensive with modern MAPF solvers (Okumura, 2024), and *(ii)* IL consistently outperforms RL (Andreychuk et al., 2025a; Veerapaneni et al., 2025).

### 2.3 MAGAT: GNN-BASED POLICY FOR MAPF

Among existing IL policies for MAPF to date, MAGAT (Li et al., 2021) is a representative approach that employs GNNs to capture pairwise agent interactions. Formally, it governs a policy $\pi_i(v \mid o_i, \mathcal{G})$ that outputs an action distribution over candidate target vertices $v$ for agent $i$, given its observation $o_i$ and the communication graph $\mathcal{G}$, which is constructed based on spatial proximity within a radius $R^{\text{comm}} \in \mathbb{N}_{>0}$. The model consists of *(i)* a CNN encoder that converts $o_i$ into the node feature $\mathbf{x}_i$, *(ii)* a GNN layer that aggregates messages from other agents following $\mathcal{G}$, and *(iii)* an MLP decoder.

Our proposed HMAGAT builds upon MAGAT. In addition to replacing the GNN with HGNN layers, HMAGAT also incorporates techniques introduced in follow-up works on MAGAT. These include stacking GNN layers (three) to enhance representational power (Veerapaneni et al., 2025) and incorporating position-based edge features embedded into $\mathcal{G}$. Our implementation of MAGAT also includes these improvements.

**Local Observation.** Throughout the rest of the paper, we fix the observation tensor $o_i$ for agent $i$ at a configuration $\mathcal{Q}$ to be centred around its current position $\mathcal{Q}[i]$, with shape $4 \times (2R^{\text{obs}} + 1) \times (2R^{\text{obs}} + 1)$, where $R^{\text{obs}} \in \mathbb{N}_{>0}$ determines the FOV size. The four channels consist of *(i)* an obstacle map, *(ii)* an agent map, *(iii)* a projection of the goal direction, and *(iv)* a normalised cost-

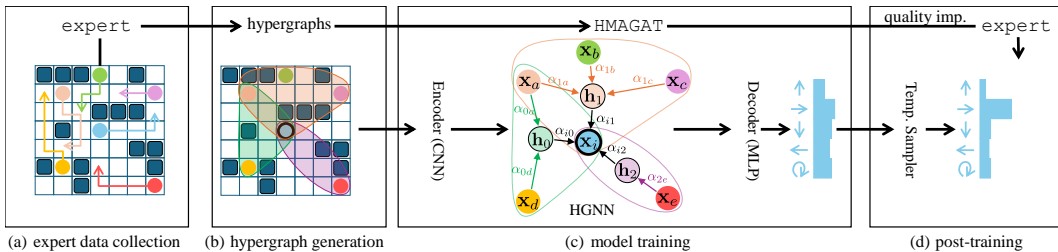

Figure 2: Overview of HMAGAT. (a) We collect demonstrations using an expert solver over 21K instances. (b) For each timestep, we extract a directed hypergraph representation (head agent shown with bold outline here). (c) These are used to train our HGNN-based model. (d) Post-training, we train the model using expert trajectories over intermediate instances, to improve solution quality. A temperature sampler is also trained to make the model more confident.

to-go map. For each location $v \in V$ within FOV, the normalised cost-to-go value is computed as $(\text{dist}(v, g_i) - \text{dist}(\mathcal{Q}[i], g_i))/(2R^{\text{obs}})$, while setting it to 1 for obstacles. This construction is commonly used in learning-based MAPF studies (Alkazzi & Okumura, 2024).

## 3  HMAGAT

While GNNs have shown promise in modelling multi-agent systems, they inherently focus on pairwise interactions between agents. This pairwise focus can limit their ability to capture the complex coupled-dynamics in multi-agent systems with interactions involving multiple agents simultaneously. We identify two key limitations of GNNs, specifically MAGAT, in this context.

**Attention in dense scenarios.** In multi-agent systems, agent interactions tend to be uneven, where for a given agent, some interactions might be more important than others. Thus, methods like MAGAT use an attentional mechanism to better model the interactions between agents. The attention scores are computed in a pairwise manner and normalised over the neighbourhood (via softmax). This, however, can be problematic in out-of-distribution dense scenarios. In such scenarios, each agent will have a dense neighbourhood, with only a few agents being truly relevant. However, since the attention scores are first computed pairwise and then normalised over the neighbourhood, the presence of many irrelevant agents will dilute the attention scores of the relevant agents. Appendix E.1 includes an informal proof that GNNs suffer from this phenomenon.

**Group Interactions.** The interactions in GNNs are inherently pairwise, while multi-agent problems, like MAPF, tend to be fundamentally a group planning problem. Models like MAGAT can assuage this to some extent by using multiple GNN layers, where deeper layers can theoretically learn to capture group interactions. However, without an explicit bias for these group interactions, the model would likely struggle to learn such interactions. We provide further details on these limitations in Appendix F.

Motivated by these observations, we propose HMAGAT, an attentional hypergraph neural network (HGNN)-based imitation learning model for MAPF. We first describe the architecture of HMAGAT, and then explore different hypergraph generation strategies.

### 3.1  ARCHITECTURE

To model the group interaction of multiple agents influencing a single agent's decision, we design directed hypergraphs, with a singleton head and a multi-node tail. We replace the GNN layers in MAGAT with our HGNN layers, resulting in a pipeline consisting of *(i)* a CNN encoder, followed by *(ii)* HGNN layers, and *(iii)* an MLP decoder.

**Communication Hypergraph.** We detail several strategies to generate a hypergraph $\mathcal{H}$ in Section 3.2. We include hyperedge features $\omega_{je}$ for each $e \in \mathcal{H}$ and $j \in T(e)$. This is set to be a

three-dimensional vector, consisting of the relative positional co-ordinates and the Manhattan distance between the agent and the hyperedge centre, which is taken to be the position of the head agent or, in the general case, the centroid of the head agents.

**Hypergraph Attention Network.** We design our HGNN in a message-passing manner, with messages going from the tail nodes to the hyperedges to the head nodes. Concretely, let $\mathbf{x}_i^{(l)}$ denote the node feature of agent $i$ at layer $l$. The layer update proceeds with:

$$\mathbf{x}_i^{(l+1)} = \sigma\left(\mathbf{W}_R^{(l)}\,\mathbf{x}_i^{(l)} + \sum_{e \in \Gamma(i) \wedge i \in H(e)} \alpha_{ie}^{(l)}\left(\mathbf{W}_h^{(l)}\,\mathbf{h}_e^{(l)}\right)\right) \tag{1}$$

$$\alpha_{ie}^{(l)} = \mathrm{softmax}\left[\mathrm{LeakyReLU}\left(\left(\mathbf{x}_i^{(l)}\right)^\top \left(\Theta_h^{(l)}\,\mathbf{h}_e^{(l)}\right)\right)\right] \tag{2}$$

$$\mathbf{h}_e^{(l)} = \sum_{j \in T(e)} \alpha_{ej}^{(l)}\left(\mathbf{W}_n^{(l)}\,\mathbf{x}_j^{(l)} + \mathbf{W}_e^{(l)}\,\mathbf{w}_{je}\right) \qquad \text{(hyperedge representation)} \tag{3}$$

$$\alpha_{ej}^{(l)} = \mathrm{softmax}\left[\mathrm{LeakyReLU}\left(\left(\sum_{i \in H(e)} \mathbf{x}_i^{(l)}/|H(e)|\right)^\top \left(\Theta_n^{(l)}\,\mathbf{x}_j^{(l)} + \Theta_e^{(l)}\,\mathbf{w}_{je}\right)\right)\right] \tag{4}$$

Here, $\mathbf{w}_{je} = \phi(\omega_{je})$ is the hyperedge feature processed via an MLP $\phi$, $\alpha_{ej}^{(l)}$ and $\alpha_{ie}^{(l)}$ are the normalised attention weights, $\mathbf{W}_{\{R,n,e,h\}}^{(l)}$ and $\Theta_{\{n,e,h\}}^{(l)}$ are the learnable weights, and $\sigma$ is a non-linearity. The above design reflects the necessity of modelling variable hypergraph and hyperedge sizes. Instead of architectures with fixed and uniquely identifiable relation types (Busbridge et al., 2019), HMAGAT is derivative from attentional HGNNs (Bai et al., 2021; Chen et al., 2020), which can accommodate dynamic and flexible hypergraph structures.

This HGNN architecture is capable of mitigating attention dilution. We provide an informal proof in Appendix E.2.

### 3.2 HYPERGRAPH GENERATION

Unlike GNNs, which naturally introduce pairwise interactions over a communication graph, defining hypergraphs for MAPF that capture group interactions is not trivial. We focus on directed hypergraphs with singleton heads and multi-node tails. An intuitive approach is to assign a

---

**Algorithm 1** COLOURINGBASEDHYPERGRAPHS

**input**: colours $C$, colouring $R$, agents $A$, positions $\mathcal{Q}$
1: $\mathcal{E} \leftarrow \emptyset$
2: **for** $v \in A, c \in C$ **do**
3: $\quad T \leftarrow \{u \in A \mid \|\mathcal{Q}[u] - \mathcal{Q}[v]\| \leq R^{\mathrm{comm}} \wedge (\mathcal{Q}[u], c) \in R\}$
4: $\quad$ **if** $T \neq \emptyset$ **then** $\mathcal{E} \leftarrow \mathcal{E} \cup \{(T \cup \{v\}, \{v\})\}$
5: **return** $(A, \mathcal{E})$ ▷ *directed hypergraph*

---

colour—potentially multiple colours—to each vertex $v \in V$, based on geometric adjacency, under the belief that agents in the same local region are more likely to interact collectively. We then form groups according to agent locations and their colours; that is, agents sharing the same colour form the tail of a hyperedge. Concretely, let $C$ be the set of colours, and let $R : V \mapsto 2^{|C|}$ denote the colouring function. Given $C$ and $R$, Alg. 1 creates a directed hypergraph subject to the constraint of a communication radius $R^{\mathrm{comm}}$. We next discuss strategies to build the colouring $R$.

**Lloyd Hypergraphs.** A principled approach to workspace partitioning is to employ a Voronoi diagram, which divides the space into regions based on distances to a set of $k$ initial seed locations. By applying Lloyd's algorithm (Lloyd, 1982; Zaman et al., 2024), one can obtain a balanced Voronoi partition with approximately equal-sized regions, regardless of the choice of initial seeds. In principle, each region can be assigned a distinct colour. How-

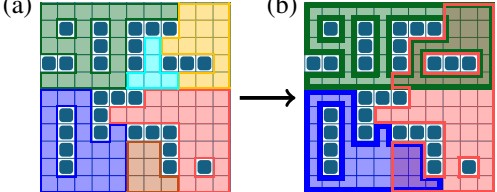

Figure 3: Going from (a) strict partitioning to (b) overlapping groups.

ever, a strict partitioning is not ideal for our setting, since agents may interact across these artificial boundaries. This motivates us to introduce "soft" borders by allowing multiple colours per vertex.

Concretely, after the Lloyd's algorithm, we discard half of the least populous colours and reassign the corresponding regions to the colours of their neighbouring regions. An illustration of this process is shown in Figure 3, with the pseudocode provided in Appendix A.

$k$-**means Hypergraphs.**    Since Lloyd's algorithm requires $O(|V|^3)$ time for updating centroids, it becomes impractical for large graphs. In a faster method, we first *diffuse* colours from $k$ randomly sampled vertices, where each vertex is associated with a $k$-dimensional vector and each dimension corresponds to one colour. Over a fixed number of iterations $T$, each vertex updates its vector to the mean of its neighbours' vectors. We then apply $k$-means clustering (MacQueen, 1967) to these vectors, followed by the "soft boundary operation" described above. This procedure reduces the complexity to $O(k|V|)$. See Appendix A for the detailed procedure.

**Shortest Distance-based Hypergraphs.**    We also explore non-colouring-based hypergraphs. Given an agent $i$, we say that two agents $j$ and $k$ should be in the tail of a hyperedge with head $i$ if agent $i$ can encounter one agent while visiting the other. We formalise this using the shortest path distances between the agents. We include further details of this strategy in Appendix A.

## 3.3    TRAINING PIPELINE

**Model Training.**    We use the POGEMA toolkit (Skrynnik et al., 2025), also employed in the development of `MAPF-GPT` (Andreychuk et al., 2025a). Following `MAPF-GPT`'s training protocol, we generate a total of 21K instances, of which, 20% feature randomly placed obstacles, while the remaining 80% are maze-like environments. Map sizes range from 17×17 to 21×21, with 16, 24, or 32 agents. For reference, `MAPF-GPT` uses 3.75M instances for training. In line with `MAPF-GPT`, we generate expert trajectories using `lacam3` (Okumura, 2024), a state-of-the-art anytime MAPF solver. We adopt a staged timeout strategy with time limits of $[1, 5, 15, 60]$s, where a longer timeout is used only if the shorter ones fail. As the training set does not include challenging situations, most instances were solved within $1$ s. HMAGAT, with $R^{comm} = 7$ and $R^{obs} = 5$, is trained on the collected expert trajectories using cross-entropy loss. We train for 200 epochs, requiring about 100 hours on an NVIDIA L40S GPU, using the AdamW optimiser (Loshchilov & Hutter, 2019).

**Online Expert.**    To mitigate the distributional shift common in imitation learning, we further apply on-demand dataset aggregation (Ross et al., 2011), collecting new trajectories where the model fails, following `MAGAT` (Li et al., 2021). To improve the quality of trajectories, after achieving 80% success rate, we check if the length of the model's solution exceeds $\delta_{buf}\times$ the expert trajectory length, with $\delta_{buf} = 1.2$. If so, we follow Andreychuk et al. (2025b) and extract instances at every $h$-th step, where $h = 16$. We then solve these instances using `lacam3` with timeouts of $[1, 2, 10]$s, adding trajectories that are $\delta_{buf}\times$ shorter than the model's trajectory. To keep the training time manageable, we limit the total number of quality improvement expert-calls to 30 per online expert phase.

**Post-Training.**    We follow `MAPF-GPT-DDG` (Andreychuk et al., 2025b) and apply a post-training phase to further improve the solution quality. This follows the same training procedure as before, but with the quality improvement online expert being called for all 500 instances and with the training consisting of $1 : 3$ ratio of the quality improvement instances and the pre-collected instances, respectively. We run the online expert after every epoch, due to the high accuracy of the model. We run this post-training for 20 epochs, requiring about 30 hours on an NVIDIA L40S GPU.

**Temperature Sampling.**    Previous works have shown that GNNs can achieve high accuracy, but tend to be miscalibrated, having low confidence in their predictions (Hsu et al., 2022; Wang et al., 2021). This is particularly problematic in MAPF, where we sample the next action based on the model's output distribution. Similar problems are likely to exist for HGNNs as well. To tackle this, we use a softmax temperature $\tau$ between $0.5$ and $1.0$. We train an RL module to dynamically adjust $\tau$ for each agent based on its local observability and action log-odds. This module is trained for 50 epochs, taking about 3 hours on an NVIDIA L40S GPU. We include further details in Appendix A.

## 4    EVALUATION

**Baselines.**    We compare HMAGAT with the following three state-of-the-art learning-based MAPF solvers: *(i)* MAGAT (Li et al., 2021), an attentional GNN-based model. *(ii)* MAPF-GPT (Andreychuk et al., 2025a), the most capable IL policy to date, using a GPT-like architecture. The authors provided three model sizes: 2M, 6M, and 85M. *(iii)* MAPF-GPT-DDG (Andreychuk et al., 2025b), a fine-tuned version of MAPF-GPT (2M) model, which the authors claim matches the 85M model. We use PIBT-based collision shielding (Okumura et al., 2022) for all these methods, as recommended by Veerapaneni et al. (2025). We also run our evaluation on SSIL (Veerapaneni et al., 2025), another GNN-based IL model, and SCRIMP (Wang et al., 2023) and DCC (Ma et al., 2021), two RL models. However, due to the vast difference in performance, we limit their results to Appendix C.

**Metrics.**    We present the success rates, the average relative sum-of-costs (Rel. SoC) with respect to lacam3 with a 30 s budget and the average runtime per-map. The success rate measures the percentage of instances in which all the agents reach their respective goals. For calculating SoC for failed instances, we follow POGEMA's (Skrynnik et al., 2025) and, in turn, MAPF-GPT's (Andreychuk et al., 2025a) strategy of assigning the episode length to be the costs for agents that fail to reach their goals. For iterative solvers, this is a valid strategy to get a lower bound on the SoC if the episode length was not limited. lacam3 is the only search-based solver evaluated, and it succeeds in all the evaluated scenarios, leading to no issues.

**Evaluation Maps.**    The commonly used MAPF-benchmarking maps can be divided into seven main categories (Stern et al., 2019): *(i)* Maze, *(ii)* Warehouse, *(iii)* Room, *(iv)* Game, *(v)* City, *(vi)* Random and *(vii)* Open.

In our evaluation below, we use *(i)* Sparse Maze and *(ii)* Empty Room with 256 step limit and 60 s time limit, *(iii)* Dense Maze, *(iv)* Dense Room and *(v)* Dense Warehouse with 512 step limit and 60 s time limit, and *(vi)* ost003d with 1024 step limit and 90 s time limit. Since Random and Open maps are just simpler versions of Maze maps, we omit their results. City maps are large and sparse, with simple obstacle structures, so we limit their evaluation to Appendix C.

### 4.1    RESULTS

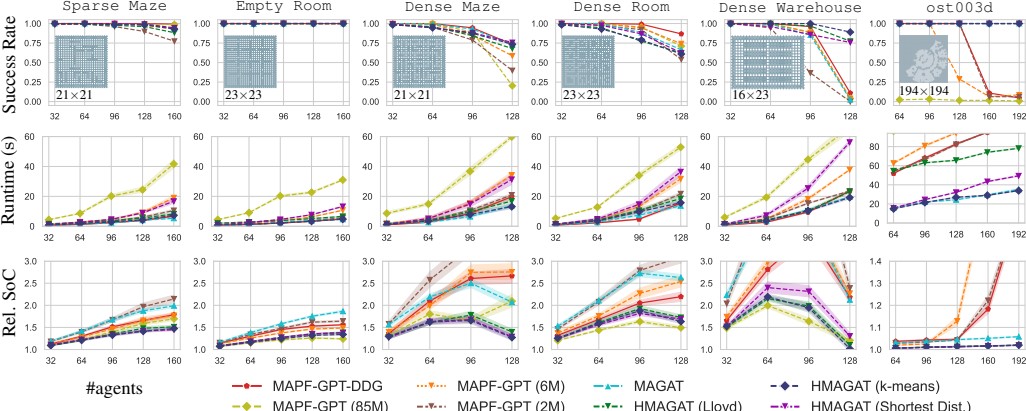

Figure 4: Evaluation of the learning-based MAPF policies, averaged over 128 instances. Transparent regions represent 95% confidence intervals. SoC is reported relative to lacam3 (w/ 30 s time limit).

Figure 4 shows the results on the different maps. We see that all HMAGAT strategies consistently obtain higher quality solutions than MAGAT, as well as larger models, like MAPF-GPT (2M), MAPF-GPT (6M) and MAPF-GPT-DDG (2M). HMAGAT remains competitive with the largest MAPF-GPT (85M) model, while being significantly faster. $k$-means and Lloyd's HMAGAT have consistently high solution quality, with the $k$-means strategy being faster, especially on the larger maps, as expected. For the rest of the analysis, we focus on $k$-means hypergraphs.

HMAGAT outperforms MAPF-GPT (85M) in the Maze maps in terms of solution quality, but MAPF-GPT (85M) outperforms HMAGAT in the Room and Warehouse maps, except for the

Table 1: Comparison of HGNN and GNN, corresponding to `HMAGAT` w/o embellishments and `MAGAT`, on different maps with the maximum agent density. Rel. SoC is relative to `lacam3`.

| Metric | Method | Dense Maze | Dense Warehouse | Dense Room | ost003d | Paris |
|---|---|---|---|---|---|---|
| Success Rate | GNN | 72.7% | 2.3% | 66.4% | 100.0% | 100.0% |
| | HGNN | 75.8% | 39.8% | 75.0% | 100.0% | 100.0% |
| Rel. SoC (95% CI) | GNN | $2.07 \pm 0.09$ | $2.12 \pm 0.10$ | $2.63 \pm 0.07$ | $1.06 \pm 0.00$ | $1.03 \pm 0.00$ |
| | HGNN | $1.79 \pm 0.09$ | $1.78 \pm 0.08$ | $2.35 \pm 0.06$ | $1.05 \pm 0.00$ | $1.02 \pm 0.00$ |

highest agent density case for `Dense Warehouse` map. In fact, `HMAGAT` achieves 75+% success rate even in this challenging scenario, while the other methods achieve <11% success rates. `MAPF-GPT` (85M) is fails to scale to `ost003d`, while `HMAGAT` obtains solutions close to `lacam3`'s quality.

**Ablation Study.** Figure 5 shows the results of our ablation study over the highest agent density scenarios on each of the small maps, where we start from `MAGAT` and incrementally add components to reach `HMAGAT`. For ease, we fix the hypergraph generation strategy to be $k$-means colouring-based. The addition of each module improves both the success rate and the solution quality, except for the RL-based temperature sampling, which trades off the success rate for improved solution quality. This shows the importance of each component for the success of `HMAGAT`.

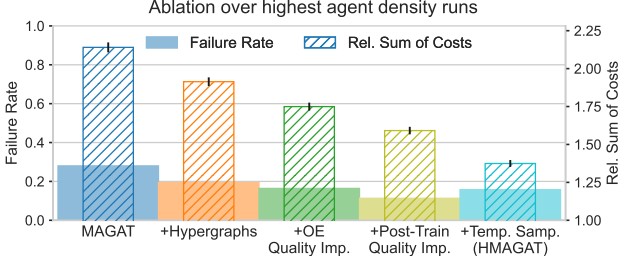

Figure 5: Ablation study. We start from `MAGAT` and incrementally add components to reach `HMAGAT`. We plot the failure rate and Rel. SoC (both the lower the better), over the highest agent density scenarios of the small maps.

## 4.2 HGNN vs GNN – Deeper Analysis

In this section, we further analyse the difference due to the use of HGNNs over GNNs. To isolate the effects and perform a direct comparison, we focus on a stripped-down version of `HMAGAT`, consisting solely of the replacement of the GNN layers with HGNNs. To emphasise the difference from `HMAGAT`, we refer to this stripped-down version as simply the HGNN model and `MAGAT` as the GNN model, for this analysis.

Table 1 shows the comparison between the two models over different maps. We see that the HGNN model consistently outperforms the GNN model across all maps, especially in the denser maps. This aligns with our hypothesis that HGNNs are better able to capture complex group interactions. The HGNN model also has a higher success rate in the dense scenarios. This gap is particularly pronounced on the `Dense Warehouse` map, where the HGNN model has a success rate of 39.8% compared to the GNN model's 2.3%.

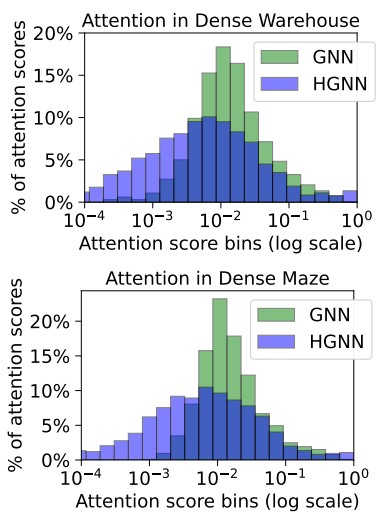

Figure 6: Attention score distribution in the first layer of the models with 128 agent instances.

**Attentional Analysis.** To better understand the difference in performance of the two models, we analyse the attention scores in the first layer of both the models. Figure 6 shows the distribution of these attention scores over a sample instance of `Dense Warehouse` and `Dense Maze` maps. The GNN model has a high concentration of attention scores in the higher-middle range, which leads to a dilution of attention, resulting in very few agents receiving high attention. On the other hand, the HGNN model has fewer agents in this range, allowing it to maintain more agents in the highest

attention score bin. This supports our hypothesis that GNNs struggle in high density scenarios due to the dilution of attention scores, which our HGNN model is able to overcome. We provide further analysis of attention dilution in Appendix H.

## 4.3 HAND-CRAFTED SCENARIOS

We now analyse the two models on hand-crafted scenarios and showcase the limitations of GNNs due to *(i)* attention dilution in dense scenarios and *(ii)* the inability to effectively capture group interactions, which HGNNs are able to overcome since they do not rely on pairwise interactions.

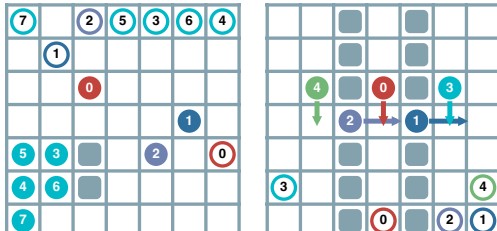

Figure 7: Hand-crafted scenarios. The filled circles represent the agent locations, while the empty ones represent their targets. **Scenario 1** (left): We form 4 groups of agents, shown by different colours here. **Scenario 2** (right): We show the next greedy actions for the agents.

Table 2: CV of attention scores (in the first layer) for agent 0 as we vary the number of agents from 4 to 8 (inclusive both) in the map described in Figure 7 (left).

| Method | Grp. 0 | Grp. 1 | Grp. 2 | Grp. 3 |
|--------|--------|--------|--------|--------|
| GNN | 29.4% | 20.0% | 14.9% | 39.2% |
| HGNN | 6.0% | 4.3% | 7.4% | 14.7% |
| Ratio | 4.94 | 4.66 | 2.01 | 2.66 |

Table 3: Percentage Shapley values for the agents with respect to agent 0 in Figure 7 (right).

| Method | Agt. 1 | Agt. 2 | Agt. 3 | Agt. 4 | Agts. 3 vs. 4 |
|--------|--------|--------|--------|--------|---------------|
| GNN | 15.0% | 60.3% | 13.1% | 11.6% | +13% |
| HGNN | 19.0% | 59.6% | 19.3% | 2.1% | +802% |

**Scenario 1.** Since, in GNNs, the attention scores are computed in a pairwise manner, a high density of agents in unimportant regions will lead to a dilution of attention. To illustrate this, we consider the scenario shown in Figure 7 (left). We can create a sequence of scenarios by starting with 4 agents (agents 0-3 in groups 0-3, respectively) and sequentially adding agents 4-7 to group 3. We focus on the first layer attention scores wrt. agent 0. By design, agent 0 primarily needs to attend to agent 2 and, potentially, agent 1, as their respective greedy paths have intersections with agent 0's. On the other hand, the agents in group 3 are not expected to interfere with agent 0's path, and thus, increasing the number of agents in group 3 should not affect the attention distribution for agent 0.

Table 2 shows the coefficient of variation (CV) of the attention scores. The GNN model exhibits a high variation in the attention scores as we increase the number of agents in less important regions, leading to a dilution of the attention scores in the important regions. In contrast, the HGNN model exhibits a much lower variation, remaining relatively unaffected by the number of agents in group 3, as desired. We include more details on the attention scores in Appendix D.

**Scenario 2.** In this scenario, we aim to illustrate the ability of HGNNs to capture group interactions as opposed to GNNs. Towards this end, we design the scenario shown in Figure 7 (right), analysing how agents 1-4 influence agent 0. To quantify the influence, we make use of Shapley values (Shapley, 1951; Lundberg & Lee, 2017), taking the mean of the absolute Shapley values for each plausible action-class log-odds prediction.

By design, we expect the following influence ranking: Agent 2 > Agent 3 ≈ Agent 1 > Agent 4. This is because agent 2's next greedy action conflicts with agent 0's. Agent 3's greedy path will want agent 1 to occupy this conflicting cell, leading to a group interaction effect and thus, a similar high influence on agent 0's actions. Since agent 2 is already moving away from agent 4, agent 4 is expected to have a low influence. From a purely pairwise perspective, agents 3 and 4 have mirrored start and goal locations wrt. agent 0, and thus, should have similar influence on agent 0's actions. However, due to group interactions, we expect agent 3 to have a higher influence than agent 4.

Table 3 shows the percentage Shapley values for the GNN and HGNN models. As expected, we see that both models assign the highest influence to agent 2. For the GNN model, agents 1, 3 and 4 have a similar influence, with agent 3 having only 13% more influence than agent 4. This highlights the inability of the GNN model to capture group interactions effectively. On the other hand, in the HGNN model, agent 3 has a significantly higher influence (802% more) than agent 4, showcasing the ability of HGNNs to capture group interactions and distinguish between the agents.

## 5 CONCLUSION

In this work, we explored MAPF as a representative multi-agent problem with complex interactions. We proposed `HMAGAT`, an MAPF solver based on hypergraph neural networks (HGNNs), as a solution to model higher order interactions. We empirically demonstrated that our approach outperforms current state-of-the-art methods, which only consider pairwise interactions. `HMAGAT` achieves superior performance to `MAPF-GPT` (85M), a $85\times$ larger model, trained on $100\times$ more data. We perform further analyses to show that, indeed, the HGNNs help to capture group interactions better than GNNs via explicit group modelling. Our results show how higher order representational learning models can lead to improved model performance, while lowering the sample complexity and model size, strongly motivating future research to explore hypergraph-based methods for challenging multi-agent problems with highly coupled interactions. As our model achieves SoTA performance with a significantly smaller parameter count than previous works, our results demonstrate that better inductive biases, such as hypergraph-based interaction modelling, are a complementary strategy to training larger models when tackling difficult multi-agent problems.

## REPRODUCIBILITY STATEMENT

Section 3 details our proposed `HMAGAT` model and training procedure. In order to make our work easily reproducible, we provide further details about the model architecture and training in Appendix A, including relevant hyperparameters. We provide our codebase as supplementary material, which includes instructions to reproduce our results as well as checkpoints for our trained models. The codebase also includes scripts to generate the instances used to train and evaluate models.

## ACKNOWLEDGMENTS

This research was funded in part by Trinity College Cambridge, European Research Council (ERC) Project 949940 (gAIa), JST ACT-X (JPMJAX22A1), and JST PRESTO (JPMJPR2513).

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

## A  FURTHER DETAILS FOR HMAGAT

### A.1  TEMPERATURE SAMPLING MODULE

We consider a lightweight temperature sampling module to predict the softmax temperature $\tau$. To train this module, we consider an actor-critic setup. For each agent, the actor takes the log-odds predicted by the model, the number of agents in its FOV, the number of obstacles in its FOV, and the distance and relative positional vector to its goal as input, passing them through a 2-layer MLP with ReLU activations, outputting a scalar $\tau \in [0.5, 1.0]$ for each agent, by applying a sigmoid activation and scaling the output. The critic takes the same input, passing them through a 2-layer MLP with ReLU and outputs a scalar value for the instance by averaging the outputs for all agents. We consider a simple reward function, where we give a reward of $+1$ to each agent if all agents reach their goals, else a reward of $-1$, in order to encourage higher success rates, while also encouraging shorter paths.

We train this module using the PPO algorithm (Schulman et al., 2017), with a clip ratio of 0.2, a discount factor of 0.99, and a GAE $\lambda$ of 0.95. We use a learning rate of $3 \times 10^{-4}$ and a batch size of 64. We train this module for 50 epochs, requiring about 3 hours on an NVIDIA L40S GPU. These are trained on 85 instances, each with 32 agents, on a $20 + 80\%$ combination of `Random` and `Maze` maps, with 15 separate instances for validation.

### A.2  HYPERGRAPH GENERATION

**Colouring-based Hypergraphs.** Algorithm 3 shows the pseudocode for generating a colouring for a given grid using Lloyd's algorithm, with overlapping regions. Algorithm 4 shows the pseudocode for generating a colouring using $k$-means clustering, with overlapping regions.

**Lloyds versus $k$-means.** Lloyd's algorithm for graphs employs Floyd-Warshall's algorithm to update centroids, which has a time complexity of $O(\sum_{c \in C} |s_c|^3)$, where $s_c$ is the size of cluster $c$. The cluster sizes can be imbalanced, leading to a worst-case time complexity of $O(|V|^3)$. In contrast, $k$-means clustering has a time complexity of $O(\sum_{c \in C} d \times |s_c|)$ for the updates, because it only

---

**Algorithm 2** SHORTESTDISTANCEBASEDHYPERGRAPHS

---

**input**: grid $G = (V, E)$, agents $A$, agent positions $\mathcal{Q}$
**params**: communication radius $R^{\text{comm}}$, dynamic slack variable $\epsilon$

1: $\mathcal{E} \leftarrow \emptyset$ ▷ *set of hyperedges*
2: **for** $v \in A$ **do**
3:    $H \leftarrow \{v\}$ ▷ *head of hyperedge*
4:    $A_C \leftarrow \{u \in A \mid \|\mathcal{Q}[u] - \mathcal{Q}[v]\| \leq R^{\text{comm}}\} \setminus \{v\}$ ▷ *communicating agents*
5:    $E_C \leftarrow \emptyset$ ▷ *initialize candidate edges*
6:    **for** $u \in A_C$ **do**
7:      **for** $w \in A_C \setminus \{u\}$ **do**
8:        **if** $\mathsf{dist}(\mathcal{Q}[v], \mathcal{Q}[u]) + \mathsf{dist}(\mathcal{Q}[u], \mathcal{Q}[w]) \leq \max(\mathsf{dist}(\mathcal{Q}[v], \mathcal{Q}[u]), \mathsf{dist}(\mathcal{Q}[v], \mathcal{Q}[w])) + \epsilon$ **then**
9:          $E_C \leftarrow E_C \cup \{(u, w), (w, u)\}$
10:    $C \leftarrow \textsc{GetCliques}(A_C, E_C)$
11:    **for** $T \in C$ **do**
12:      $\mathcal{E} \leftarrow \mathcal{E} \cup \{(T \cup \{v\}, H)\}$
13: **return** $\mathcal{E}$

---

**Algorithm 3** GRIDCOLOURING

---

**input**: grid $G = (V, E)$
**params**: number of initial colours $k'$, number of final colours $k$, number of iterations $n$

1: $R', C' \leftarrow \textsc{LloydsOnGraph}(G, k', n)$ ▷ *Rigid colouring from Lloyd's*
2: $C \leftarrow k$-most populous colours in $R'$ ▷ *We use $k = k'/2$*
3: $R \leftarrow \{(v, c) \in R' \mid c \in C\}$ ▷ *Discard least populous colourings*
4: $U \leftarrow V \setminus \{v \mid \exists c \in C.(v, c) \in R\}$ ▷ *Initial uncoloured vertices*
5: **while** $\exists v \in V . !\exists c \in C . (v, c) \in R$ **do** ▷ *While there are uncoloured vertices*
6:    $R_{\text{old}} \leftarrow R$
7:    **for** $u \in U$ **do**
8:      $R \leftarrow R \cup \{(u, c) \in V \times C \mid \exists v \in \mathsf{neigh}(u) . (v, c) \in R_{\text{old}}\}$ ▷ *Take neighbours' colours*
9: **return** $R, C$ ▷ *Less rigid colouring*

---

requires computing the mean of the features of the vertices in the cluster, where $d$ is the feature dimension. Since, in our case, $d = k$ (the number of clusters) and $\sum_{c \in C} |s_c| = |V|$, this complexity becomes $O(k \times |V|)$.

With a moderate graph size, this cost is not prohibitive with respect to the model inference. E.g., on the Dense Warehouse map with 128 agents, HMAGAT takes on average 19s to solve, out of which 0.04s are spent on the colouring of the map. 0.005s ( 8% of inference time) are spent on constructing the hypergraphs from the colouring at each time step. In practice, this can be highly optimized using C++.

**Shortest Distance-based Hypergraphs.** Algorithm 2 shows the pseudocode for generating shortest distance-based hypergraphs. In order to get faster clique computations, we simply sample cliques by randomly selecting an agent from the communicating agents, and then greedily adding agents to the clique by checking if the new agent is connected to all the existing agents in the clique. We repeat this process until we have each communicating agent in at least one clique, each time selecting one of the non-selected agents as the starting agent. We repeat the process if we have less than 5 total cliques, in order to ensure that we have a reasonable number of sampled cliques even in highly connected distance graphs. To lower the computation time, we only generate new hypergraphs every 5 timesteps.

## B RADAR PLOT DETAILS

Figure 1c shows the performance of various learning-based MAPF solvers on the small maps, i.e. on `Sparse Maze`, `Empty Room`, `Sparse Warehouse`, `Dense Maze`, `Dense Room` and `Dense Warehouse`. More details on the maps can be found in Appendix C. Figure 8 shows more detailed results on these maps. We define the metrics used in the radar plot as follows:

$$\text{Sol. Quality} = \begin{cases} \text{SoC}_{\text{best}}/\text{SoC} \\ 0 \end{cases} \quad \text{, if no solution found} \tag{5}$$

---

**Algorithm 4** KMEANSCOLOURING

---

**input**: grid $G = (V, E)$
**params**: number of initial colours $k'$, number of final colours $k$, number of iterations $n$

1:   $X \leftarrow \mathbf{0}_{|V| \times k'}$          ▷ *Initial colour vectors*
2:   $X \leftarrow$ Randomly assign $k'$ vertices to unique colours (i.e. set $X[v, c] := 1$)     ▷ *Random seeding*
3:   **for** $i \in [1, n]$ **do**         ▷ *Diffuse colours*
4:      $X_{\text{old}} \leftarrow X$
5:      **for** $v \in V$ **do**
6:          $X[v] \leftarrow \sum_{u \in \mathsf{neigh}(v)} X_{\text{old}}[u]$
7:          $X[v] \leftarrow X[v] / \|X[v]\|_1$         ▷ *Mean of neighbours' colours*
8:   $R', C' \leftarrow$ KMEANS$(X, k', n)$         ▷ *Rigid colouring from k-means*
9:   $C \leftarrow k$-most populous colours in $R'$         ▷ *We use $k = k'/2$*
10:   $R \leftarrow \{(v, c) \in R' \mid c \in C\}$         ▷ *Discard least populous colourings*
11:   $U \leftarrow V \setminus \{v \mid \exists c \in C.(v, c) \in R\}$         ▷ *Initial uncoloured vertices*
12:   **while** $\exists v \in V . !\exists c \in C . (v, c) \in R$ **do**         ▷ *While there are uncoloured vertices*
13:      $R_{\text{old}} \leftarrow R$
14:      **for** $u \in U$ **do**
15:          $R \leftarrow R \cup \{(u, c) \in V \times C \mid \exists v \in \mathsf{neigh}(u) . (v, c) \in R_{\text{old}}\}$     ▷ *Take neighbours' colours*
16:   **return** $R, C$         ▷ *Less rigid colouring*

---

Table 4: Map details.

| Map | Size (H × W) | Obstacle Density | Step Limit | Time Limit |
|---|---|---|---|---|
| Sparse Maze | $21 \times 21$ | $8 - 24\%$ | 256 | 60 s |
| Empty Room | $23 \times 23$ | ~20% | 256 | 60 s |
| Sparse Warehouse | $22 \times 23$ | ~32% | 256 | 60 s |
| Dense Maze | $21 \times 21$ | $30 - 40\%$ | 512 | 60 s |
| Dense Room | $23 \times 23$ | $34 - 42\%$ | 512 | 60 s |
| Dense Warehouse | $16 \times 23$ | ~43% | 512 | 60 s |
| ost003d | $194 \times 194$ | ~65% | 1024 | 90 s |
| lak303d | $194 \times 194$ | ~61% | 1024 | 90 s |
| Paris | $256 \times 256$ | ~28% | 1024 | 90 s |
| Berlin | $256 \times 256$ | ~27% | 1024 | 90 s |

$$\text{Scalability} = \begin{cases} \frac{\text{runtime(\#agents)}}{\#\text{agents}} \cdot \frac{\text{Min. \#agents}}{\text{runtime(Min. \#agents)}} \\ 0 \qquad\qquad\qquad\qquad\qquad\qquad\quad , \text{if timed out} \end{cases} \tag{6}$$

## C  FURTHER EVALUATION RESULTS

### C.1  MORE EVALUATION MAPS

We provide further evaluation results on more maps with more methods here. Table 4 shows the details for each map. ost003d and lak303d are Game maps, while Paris and Berlin are City maps, all taken from the MovingAI pathfinding repository (Sturtevant, 2012).

### C.2  COMPARISON OF HMAGAT VARIANTS

**Small maps.** Figure 8 shows the results on the small maps. On the sparse maps, all HMAGAT variants perform nearly identically in terms of SoC. They also have similar success rates, except for $k$-means-based HMAGAT on Sparse Warehouse, where it has slightly worse success rate. $k$-means-based and Lloyds-based HMAGAT have similar runtimes, while the shortest-distance based HMAGAT is slightly slower, with the gap being more pronounced the more agents there are. This is expected because unlike the colouring-based methods that compute the colouring irrespective of the agent configurations, the shortest-distance based method's time complexity increases with the number of agents.

On the Dense Maze and Dense Room maps, all HMAGAT variants again have similar SoC and success rates. However, on Dense Warehouse, the shortest-distance based HMAGAT has a worse

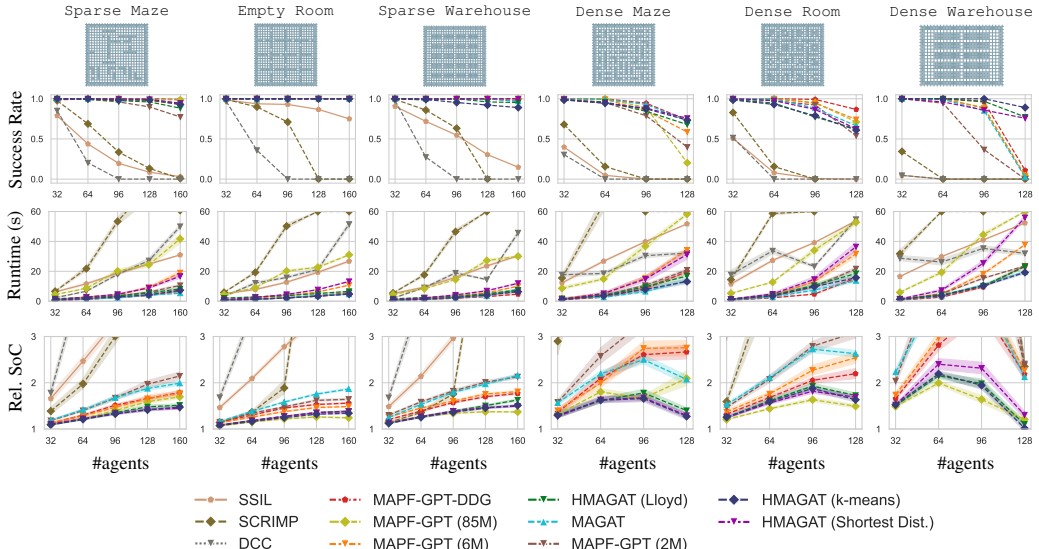

Figure 8: Evaluation of the learning-based MAPF policies on the small maps, averaged over 128 instances. Transparent regions represent 95% confidence intervals. SoC is reported relative to `lacam3` (w/ 30 s time limit).

SoC, and both the shortest-distance based and Lloyds-based `HMAGAT` have slightly worse success rates. In these scenarios, the shortest-distance based `HMAGAT` is again the slowest, with the gap being much more pronounced. This is expected because unlike the colouring-based methods that only need to compute the colouring once, the shortest-distance based method needs to compute the shortest distances every 5 timesteps, which means that for the more complex maps, where the agents take longer to reach their goals, the overhead of computing the shortest distances is more pronounced.

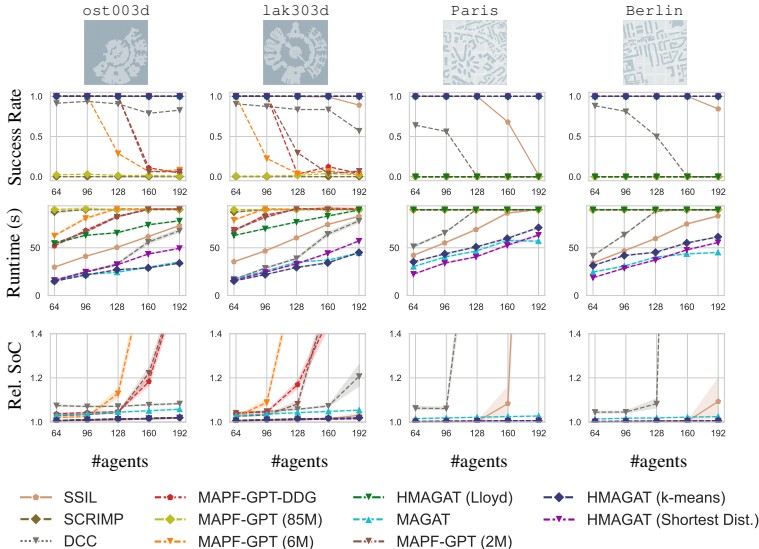

Figure 9: Evaluation of the learning-based MAPF policies on the large maps, averaged over 128 instances. Transparent regions represent 95% confidence intervals. SoC is reported relative to `lacam3` (w/ 30 s time limit).

**Large maps.** Figure 9 shows the results on the large maps. On the `Game` maps, again all `HMAGAT` variants perform nearly identically in terms of SoC and all have perfect success rates. $k$-means-based `HMAGAT` is the fastest, with the shortest-distance based `HMAGAT` being slightly slower, with the gap increasing with the number of agents. Lloyds-based `HMAGAT` is the slowest, although the gap does not increase with the number of agents. This is expected because the colouring is computed only once and is independent of the number of agents. Lloyds is slower than $k$-means as noted in Appendix A and this becomes apparent here, where $|V|$ is large.

On the `City` maps, $k$-means-based and shortest-distance based `HMAGAT` perform nearly identically in terms of SoC and have perfect success rates. Lloyds-based `HMAGAT`, on the other hand, times out in these scenarios. This is because $|V|$ here is even larger than the `Game` maps. We see that the shortest-distance based `HMAGAT` is faster than $k$-means-based `HMAGAT` here, although the gap decreases with the number of agents. This is expected because the total time complexity of hypergraph generation in shortest-distance based method depends on the number of time steps and the number of agents, while for the $k$-means-based method, it depends on $|V|$. Since we are dealing with a large and sparse map, $|V|$ is large leading to a higher time complexity for the $k$-means-based method, but not for the shortest-distance based method, which is independent of $|V|$ (assuming the number of time steps as an independent variable). However, as the number of agents increases, the time complexity of the shortest-distance based method increases, leading to a decrease in the gap.

We see that $k$-means-based `HMAGAT` obtains a good balance between performance and speed across all maps, and so, for the rest of the analysis, we focus on this variant.

### C.3 COMPARISON OF `HMAGAT` WITH OTHER METHODS

**Small maps.** Figure 8 shows the results on the small maps. `HMAGAT` consistently outperforms `MAGAT`, `MAPF-GPT-DDG`, `MAPF-GPT` (6M), `MAPF-GPT` (2M), `SSIL`, `SCRIMP` and `DCC` in terms of SoC and success rates across all maps and agent counts. The only exception being `MAPF-GPT-DDG` and `MAPF-GPT` on `Dense Room`, where they have higher success rates, but still have worse SoC.

On the sparse maps, `HMAGAT` and `MAPF-GPT` (85M) achieve similar SoC, with `HMAGAT` having slightly worse performance on the highest agent count scenario for `Empty Room` and `Sparse Warehouse`, but better performance on `Sparse Maze`. On the `Dense Maze` map, `HMAGAT` consistently outperforms `MAPF-GPT` (85M), with the gap increasing a fair amount for the highest agent count scenario. For the `Dense Room` and `Dense Warehouse` maps, `MAPF-GPT` (85M) has a slightly better SoC, except for the highest agent count scenario on `Dense Warehouse`, where `HMAGAT` has a better SoC. In this scenario, `HMAGAT` has a significantly higher success rate, achieving $89.1\%$ success rate compared to $4\%$ for `MAPF-GPT` (85M).

Notably, `HMAGAT` is significantly faster than `MAPF-GPT` (85M) across all maps and agent counts, and has a lower slope in the runtime vs. number of agents plots.

**Large maps.** Figure 9 shows the results on the large maps. `HMAGAT` consistently outperforms `MAGAT`, `MAPF-GPT` (85M), `MAPF-GPT-DDG`, `MAPF-GPT` (6M), `MAPF-GPT` (2M), `SCRIMP` and `DCC`. `HMAGAT` and `MAGAT` are the only two methods that are able to easily scale to these maps and have perfect success rates across all maps and agent counts. `SSIL` achieves a similar SoC to `HMAGAT` for the lower agent count scenarios, but seems unable to scale to various higher agent count scenarios.

`HMAGAT` consistently achieves a Rel. SoC close to 1.0, i.e. it produces solutions close to `lacam3`'s quality.

**MAPF-GPT (85M) and MAPF-GPT-DDG.** Andreychuk et al. (2025b) claim that `MAPF-GPT-DDG` matches the performance of `MAPF-GPT` (85M), and, at times, even outperforms it. However, in our experiments, we see that `MAPF-GPT-DDG` consistently underperforms `MAPF-GPT` (85M), except for the large maps, where `MAPF-GPT` (85M) fails to scale. We believe that this discrepancy is due to the fact that Andreychuk et al. (2025b) evaluate these models without PIBT-based collision shielding (Okumura et al., 2022; Veerapaneni et al., 2024). `MAPF-GPT-DDG` likely produces solutions with lower collisions, due to the finetuning, leading to a better performance than `MAPF-GPT` (85M) when not using any advanced collision shielding. To

verify this, we evaluated `MAPF-GPT-DDG` and `MAPF-GPT` (85M) without PIBT-based collision shielding on `Sparse Maze`, `Empty Room` and `Sparse Warehouse` maps with 128 agents each. Table 5 shows the results. We see that `MAPF-GPT` (85M) struggles even on these sparse maps and `MAPF-GPT-DDG` consistently outperforms it, verifying our hypothesis.

However, as seen in Figure 8, when using PIBT-based collision shielding, `MAPF-GPT` (85M) consistently outperforms `MAPF-GPT-DDG`.

Table 5: Comparison of `MAPF-GPT` (85M) and `MAPF-GPT-DDG` without PIBT-based collision shielding. Averaged over 128 instances, each with 128 agents.

| Metric | Method | Sparse Maze | Empty Room | Sparse Warehouse |
|---|---|---|---|---|
| Success Rate | MAPF-GPT (85M) | 6.3% | 2.3% | 39.1% |
| | MAPF-GPT-DDG | 76.6% | 96.9% | 99.2% |
| Rel. SoC (95% CI) | MAPF-GPT (85M) | $6.93 \pm 0.24$ | $5.81 \pm 0.14$ | $5.61 \pm 0.23$ |
| | MAPF-GPT-DDG | $3.81 \pm 0.20$ | $3.02 \pm 0.14$ | $3.05 \pm 0.11$ |

### C.4 HGNN VS GNN ON MAXIMUM AGENT DENSITY

Table 6 shows the performance of HGNN and GNN on different maps with the maximum agent density with additional maps compared to Table 1. We see that HGNN consistently outperforms GNN in terms of solution quality.

## D FURTHER DETAILS ON HAND CRAFTED SCENARIOS

**Scenario 1.** To calculate $a_{i,j}^{(l)}$ the attention a node $i$ pays to an agent $j$ in the $l$-th layer of `HMAGAT`, we simply use the following sum:

$$a_{i,j}^{(l)} = \sum_{e \in \Gamma(i) \land i \in H(e) \land j \in T(e)} \alpha_{ie}^{(l)} \alpha_{ej}^{(l)} \tag{7}$$

We present the attention scores for the GNN and HGNN models in the first layer Tables 7 and 8. We also present the attention scores averaged across all layers in Tables 9 and 10, with the coefficient of variation in Table 11. We see that the HGNN model consistently has a lower coefficient of variation across all groups, indicating that it is more robust to the increase in the number of agents in low-importance regions.

We have seen that the HGNN model is able to handle the increase in the number of agents in less important regions. However, we also need to ensure that when the number of agents in the important regions increases, we do not see a significant decrease in the attention scores for this region. Towards this end, we construct a scenario as shown in Figure 10.

In contrast to the scenario described in Figure 7 (left), we now have agent 0's greedy path overlapping with paths of agents in group 3, i.e the group for which we vary the number of agents. Thus, agent 0 is expected to primarily attend to agents in group 3. Table 12 shows the HGNN attention scores for this scenario. We see that the attention scores for group 3 stay high as we vary the number of agents, indicating that the HGNN model is able to maintain high attention scores for the important region.

**Scenario 2.** While calculating the Shapley values, we average the scores between the ones computed on the original map shown in Figure 7 (right) and map formed by horizontally flipping it, in order to reduce the effect of any asymmetries learnt by the models.

## E INFORMAL PROOF OF ATTENTION DILUTION

We provide an informal analysis, demonstrating that GNNs face attention dilution, which HGNNs are capable of mitigating. We first start with demonstrating that GNNs suffer from attention dilution.

Table 6: Comparison of HGNN and GNN, corresponding to `HMAGAT` w/o embellishments and `MAGAT`, on different maps with the maximum agent density. Results are over 128 instances per map. Rel. SoC is the sum-of-costs relative to `lacam3`, for which we show the 95% confidence intervals.

| Metric | Method | Sparse Maze | Sparse Warehouse | Sparse Room | ost003d | lak303d |
|---|---|---|---|---|---|---|
| Success Rate | GNN | 93.8% | 96.9% | 100.0% | 100.0% | 100.0% |
| | HGNN | 93.0% | 100.0% | 100.0% | 100.0% | 100.0% |
| Rel. SoC | GNN | $1.99 \pm 0.04$ | $2.16 \pm 0.04$ | $1.86 \pm 0.02$ | $1.06 \pm 0.00$ | $1.05 \pm 0.00$ |
| | HGNN | $1.91 \pm 0.04$ | $1.95 \pm 0.02$ | $1.69 \pm 0.02$ | $1.05 \pm 0.00$ | $1.04 \pm 0.00$ |

| Metric | Method | Dense Maze | Dense Warehouse | Dense Room | Paris | Berlin |
|---|---|---|---|---|---|---|
| Success Rate | GNN | 72.7% | 2.3% | 66.4% | 100.0% | 100.0% |
| | HGNN | 75.8% | 39.8% | 75.0% | 100.0% | 100.0% |
| Rel. SoC | GNN | $2.07 \pm 0.09$ | $2.12 \pm 0.10$ | $2.63 \pm 0.07$ | $1.03 \pm 0.00$ | $1.02 \pm 0.00$ |
| | HGNN | $1.79 \pm 0.09$ | $1.78 \pm 0.08$ | $2.35 \pm 0.06$ | $1.02 \pm 0.00$ | $1.02 \pm 0.00$ |

Table 7: GNN Attention scores in the first layer for agent 0 over the different groups in Figure 7 (left).

| # Agents | Group 0 | Group 1 | Group 2 | Group 3 |
|---|---|---|---|---|
| 4 Agents | 0.394750 | 0.257375 | 0.192023 | 0.155852 |
| 5 Agents | 0.286059 | 0.234413 | 0.186970 | 0.292558 |
| 6 Agents | 0.251166 | 0.202690 | 0.161970 | 0.384175 |
| 7 Agents | 0.215447 | 0.172166 | 0.146812 | 0.465576 |
| 8 Agents | 0.194606 | 0.159428 | 0.135639 | 0.510326 |

Table 8: HGNN Attention scores in the first layer for agent 0 over the different groups in Figure 7 (left).

| # Agents | Group 0 | Group 1 | Group 2 | Group 3 |
|---|---|---|---|---|
| 4 Agents | 0.024621 | 0.104523 | 0.493603 | 0.377252 |
| 5 Agents | 0.027046 | 0.110814 | 0.583928 | 0.278213 |
| 6 Agents | 0.024019 | 0.113196 | 0.603738 | 0.259047 |
| 7 Agents | 0.023478 | 0.102690 | 0.571674 | 0.302157 |
| 8 Agents | 0.023619 | 0.104273 | 0.564101 | 0.308007 |

Table 9: GNN Attention scores averaged across layers for agent 0 over the different groups in Figure 7 (left).

| # Agents | Group 0 | Group 1 | Group 2 | Group 3 |
|---|---|---|---|---|
| 4 Agents | 0.170300 | 0.381573 | 0.190819 | 0.257309 |
| 5 Agents | 0.130953 | 0.349720 | 0.188699 | 0.330629 |
| 6 Agents | 0.117691 | 0.329456 | 0.171886 | 0.380967 |
| 7 Agents | 0.109234 | 0.278605 | 0.164050 | 0.448111 |
| 8 Agents | 0.099965 | 0.249049 | 0.156532 | 0.494454 |

Table 10: HGNN Attention scores averaged across layers for agent 0 over the different groups in Figure 7 (left).

| # Agents | Group 0 | Group 1 | Group 2 | Group 3 |
|---|---|---|---|---|
| 4 Agents | 0.103088 | 0.274550 | 0.305216 | 0.317146 |
| 5 Agents | 0.119637 | 0.247633 | 0.350739 | 0.281991 |
| 6 Agents | 0.118572 | 0.239537 | 0.334259 | 0.307632 |
| 7 Agents | 0.120345 | 0.219097 | 0.329250 | 0.331308 |
| 8 Agents | 0.117992 | 0.221249 | 0.319412 | 0.341347 |

Table 11: Coefficient of variation of the average-attention scores (across layers) when varying the number of agents in group 3 in Figure 7 (left).

| Method | Group 0 | Group 1 | Group 2 | Group 3 |
|---|---|---|---|---|
| GNN | 21.85% | 16.87% | 8.63% | 24.54% |
| HGNN | 6.24% | 9.39% | 5.17% | 7.26% |
| GNN / HGNN | 3.5012 | 1.7957 | 1.6686 | 3.3794 |

### E.1 GNNs suffer from attention dilution

Consider a graph $\mathcal{G} = (\mathcal{V}, \mathcal{E})$ and a node $i \in \mathcal{V}$ with neighbourhood $\mathcal{N}_i$, where we can partition $\mathcal{N}_i$ into two subsets:

- $\mathcal{N}_i^{\text{noise}}$: nodes in a similar locality that do not aid in predicting $i$'s label.
- $\mathcal{N}_i^{\text{rest}}$: the remaining nodes in the neighbourhood.

Let $a_{ij}$ denote the pre-softmax attention score for node $j \in \mathcal{N}_i$ with respect to node $i$, and let the normalised attention scores be given by $\alpha_{ij} = \frac{\exp(a_{ij})}{\sum_{k \in \mathcal{N}_i} \exp(a_{ik})}$.

To demonstrate attention dilution, we construct a new graph $\mathcal{G}' = (\mathcal{V}', \mathcal{E}')$ by adding more noisy nodes to graph $\mathcal{G}$. Specifically, we define the neighbourhood of node $i$ in the new graph as $\mathcal{N}_i' = \mathcal{N}_i \cup A'$, where nodes in $A'$ are in a similar locality as $\mathcal{N}_i^{\text{noise}}$. The other nodes retain their original features. Let $\mathbf{x}_j'$ and $\omega_{ij}'$ represent the node embeddings and edge vectors in the new graph, with $a_{ij}'$ and $\alpha_{ij}'$ denoting the pre-softmax and normalised attention scores, respectively.

Since, $\mathbf{x}_j' = \mathbf{x}_j$ and $\omega_{ij}' = \omega_{ij}$ for all $j \in \mathcal{N}_i$, we have $a_{ij}' = a_{ij}$ for all $j \in \mathcal{N}_i$.

Figure 10: Hand-crafted scenario to illustrate the effect of increasing number of agents in important regions on the attention scores. Filled circles represent the agent locations, while the empty ones represent their targets. Agents are grouped via colours, with groups 0 to 2 consisting of their corresponding agents only, while group 3 consists of agents 3 to 7. We start with the agents numbered from 0 to 3, and add more agents to group 3, in order of their numbering.

Table 12: HGNN Attention scores in the first layer over groups.

| # Agents | Group 0 | Group 1 | Group 2 | Group 3 |
|----------|---------|---------|---------|---------|
| 4 Agents | 0.000949 | 0.001296 | 0.001605 | 0.996150 |
| 5 Agents | 0.002126 | 0.000187 | 0.000252 | 0.997436 |
| 6 Agents | 0.002358 | 0.000235 | 0.000335 | 0.997073 |
| 7 Agents | 0.002095 | 0.002070 | 0.002800 | 0.993035 |
| 8 Agents | 0.005777 | 0.003658 | 0.005720 | 0.984845 |

Thus, for all nodes $j \in \mathcal{N}_i^{\text{rest}}$, we have:

$$\alpha'_{ij} = \frac{\exp(a'_{ij})}{\sum_{k \in \mathcal{N}'_i} \exp(a'_{ik})} = \frac{\exp(a_{ij})}{\sum_{k \in \mathcal{N}_i} \exp(a_{ik}) + \sum_{k \in A'} \exp(a'_{ik})} < \alpha_{ij}$$

This shows that the attention scores for nodes in $\mathcal{N}_i^{\text{rest}}$ have decreased due to the addition of noisy nodes in $A'$, demonstrating attention dilution.

### E.2 HGNNs CAN MITIGATE ATTENTION DILUTION

Now, we demonstrate that HGNNs can mitigate attention dilution:

Consider a hypergraph $\mathcal{H} = (\mathcal{V}, \mathcal{E}_H)$ corresponding to the same instance represented by the graph $\mathcal{G}$. Again, let node $i \in \mathcal{V}$ have a node neighbourhood that can be partitioned into two subsets – $\mathcal{N}_i^{\text{noise}}$ and $\mathcal{N}_i^{\text{rest}}$, with nodes in $\mathcal{N}_i^{\text{noise}}$ being in a similar locality. Assume an appropriate hypergraph structure such that nodes in $\mathcal{N}_i^{\text{noise}}$ belong to the tail of a single hyperedge $e_\sigma \in \mathcal{E}_H$, while nodes in $\mathcal{N}_i^{\text{rest}}$ belong to the tails of other hyperedges, $\mathcal{E}_{\text{rest}} \subset \mathcal{E}_H$, with $i$ being in the head of all these hyperedges. Let $\alpha_{ej}$ be the attention score for node $j$ with respect to hyperedge $e$, and let $a_{ie}$ and $\alpha_{ie}$ be the pre-softmax and normalized attention scores for hyperedge $e$ with respect to node $i$, respectively. Here we have:

$$\alpha_{ie} = \frac{\exp(a_{ij})}{\sum_{k \in \{e_\sigma\} \cup \mathcal{E}_{\text{rest}}} \exp(a_{ik})}$$

Also, let the hyperedge embedding for an hyperedge $e$ be $\mathbf{h}_e$.

Now, we construct the new hypergraph $\mathcal{H}' = (\mathcal{V}', \mathcal{E}'_H)$, corresponding to the same instance represented by the graph $\mathcal{G}'$. Here we have $\mathcal{V}' = \mathcal{V} \cup A'$, with nodes in $A'$ having similar locality to nodes in $\mathcal{N}_i^{\text{noise}}$. Since they share a locality, we can assume that nodes in $\mathcal{N}_i^{\text{noise}} \cup A'$ belong to the tail of the same hyperedge $e'_\sigma \in \mathcal{E}'_H$, while nodes in $\mathcal{N}_i^{\text{rest}}$ belong to the tails of other hyperedges, $\mathcal{E}_{\text{rest}}$, with $i$ being in the head of all the hyperedges, similar to the original hypergraph.

For all nodes $j \in \mathcal{N}_i^{\text{rest}}$, we have:

$$\alpha'_{ej} = \alpha_{ej}$$

This is because the hyperedges in $\mathcal{E}_{\text{rest}}$ have the same tail nodes as in the original hypergraph, leading to the same attention scores. This means the hyperedge embeddings for hyperedges in $\mathcal{E}_{\text{rest}}$ remain unchanged, i.e. $\mathbf{h}'_e = \mathbf{h}_e$ for all $e \in \mathcal{E}_{\text{rest}}$. This also means that the pre-softmax attention scores for hyperedges in $\mathcal{E}_{\text{rest}}$ remain unchanged, i.e. $a'_{ie} = a_{ie}$ for all $e \in \mathcal{E}_{\text{rest}}$.

With regards to hyperedge $e'_\sigma$: The original hyperedge $e_\sigma$ consisted of nodes not helpful with $i$'s label prediction. Since the hyperedge embeddings are computed as a weighted combination of the node embeddings, simply adding more nodes to the tail of the hyperedge does not increase the magnitude or relevance of the hyperedge embedding. Since the new nodes, $A'$, are also not helpful for predicting node $i$'s label, they will not increase the hyperedge's attention score. Thus, we can assume that the pre-softmax attention score for hyperedge $e'_\sigma$ remains unchanged, i.e. $a'_{ie'_\sigma} = a_{ie_\sigma}$.

Thus, for all nodes $j \in \mathcal{N}_i^{\text{rest}}$, we have:

$$\alpha'_{ij} = \frac{\exp(a'_{ij})}{\sum_{k \in \{e'_\sigma\} \cup \mathcal{E}_{\text{rest}}} \exp(a'_{ik})} = \frac{\exp(a_{ij})}{\sum_{k \in \{e_\sigma\} \cup \mathcal{E}_{\text{rest}}} \exp(a_{ik})} = \alpha_{ij}$$

This shows that the attention scores for nodes in $\mathcal{N}_i^{\text{rest}}$ remain unchanged despite the addition of noisy nodes in $A'$, demonstrating that HGNNs can mitigate attention dilution. Thus, unlike in GNNs, introducing additional noisy nodes does not dilute the attention assigned to informative nodes.

## F    FURTHER DETAILS ON PAIRWISE INTERACTION MODELLING FOR MAPF

MAPF solvers take as input information about each agent, and use this information to compute paths for all agents. For MAGAT/GNNs, each agent's information is represented using relevant node embeddings, while for MAPF-GPT/transformers, this information is represented using relevant token embeddings. An action for agent $i$ is predicted based on the information of agent $i$ and the information of other agents. When only pairwise interactions are considered, we say the action for agent $i$, $o_i$, is predicted as follows:

$$o_i = \bigoplus_{j \in \mathcal{N}_i} \phi(\mathbf{x}_i, \mathbf{x}_j)$$

where $\mathbf{x}_i$ is the information of agent $i$, $\phi$ is a function that models the interaction between agent $i$ and agent $j$ and $\bigoplus$ is an aggregation function for the interactions.

In the context of Figure 1(c), we show the pairwise interactions, $\phi$, in parts (iii-v), with respect to agent A. $\bigoplus$ can be taken to be any aggregation function; in this case, it could just be the intersection operation among the pairwise paths. This results in the predicted path for the agents as shown in part (ii). However, we see that this predicted path is not optimal, and the purely pairwise interactions fail to capture the group interaction that leads to the optimal solution shown in part (i).

In the context of MAGAT and MAPF-GPT, $\phi(\mathbf{x}_i, \mathbf{x}_j)$ is the messages in the message passing framework and the value vector in the attention layer, respectively, while $\bigoplus$ is the attention-based weighted-mean aggregation in both cases. Although these mechanisms are only using pairwise interactions, they can still perform well on MAPF tasks, by learning to approximate group interactions. They can do so by: *(i)* using high-dimensional embeddings that can simply encode all the information about other agents, followed by a complex enough function to decode this information, or *(ii)* stacking multiple layers of pairwise interactions, which can allow for indirect capture of group interactions. Both MAGAT and MAPF-GPT use a combination of these two strategies. However, both of these strategies have their limitations.

The first strategy relies on the model's ability to encode and decode all relevant information about other agents within high-dimensional embeddings. This may work well for small-scale problems, but as the number of agents increases, the amount of information that needs to be encoded grows significantly, making it increasingly difficult for the model to effectively capture all necessary interactions. The second strategy, stacking multiple layers of pairwise interactions, can help indirectly capture group interactions, but given a fixed number of layers, there is a limit to the complexity of group interactions that can be effectively modelled. As the number of agents increases, the depth required to capture all relevant group interactions may exceed practical limits.

In contrast, hypergraph-based approaches can directly model the group interactions among agents, allowing for a more natural and efficient representation of the MAPF problem. Also, purely from an alignment perspective, hypergraph-based approaches align better with the underlying structure of interactions in optimal MAPF solutions.

## G    FAILURE MODE ANALYSIS OF HMAGAT

In Figure 4, we see that HMAGAT ($k$-means) is a lower success rate on Dense Room with 128 agents compared to MAPF-GPT-DDG. Below, we analyse the failure modes of HMAGAT in these scenarios.

First, we note that even in scenarios where HMAGAT fails in the Dense Room maps, it achieves an average of 97.3% partial success rate (percentage of agents that reach their goals at the end of task). Of the 2.7% agents that failed to reach their goals at the final timestep, 90% reached their goal sometime during the programme execution, but later had to move.

Table 13 shows the breakdown of the failure modes for these 2.7% of the agents that fail to reach their goals. Here, we say an agent was deadlocked if, for the last 5 actions, it stayed at the same

Table 13: Failure mode analysis of HMAGAT ($k$-means) on Dense Room with 128 agents.

| Failure Mode | Percentage |
|---|---|
| Deadlock | 3.4% |
| Livelock | 46.4% |
| Hit max timestep limit | 50.2% |

location. An agent was said to be livelocked, if it only oscillated between locations at least 3 times before hitting the max timestep limit. Rest of the agents were assumed to have failed due to simply hitting the max timestep limit.

We performed further analysis on the livelocked agents, and found that all livelocks involved oscillation between only two locations. Given this, it should be possible to improve the success rate of HMAGAT, using some straightforward deadlock/livelock detection mechanism, escaping the deadlock/livelock by increasing the sampling temperature or following a greedy path whenever a deadlock/livelock is detected.

## H    FURTHER ATTENTION DILUTION EXPERIMENTS

In Figure 6, we have seen how GNNs suffer from attention dilution, and HGNNs can mitigate them on single instances of Dense Maze and Dense Warehouse maps.

In order to further verify this, we perform experiments across multiple instances of highest agent density scenarios across different maps. As a relative measure of the attention dilution, we calculate the average normalised entropy of the attention scores, averaged across all execution steps and 128 instances, using the formula below:

$$H = -\sum_{i \in \mathcal{V}} \frac{1}{|\mathcal{V}|} \left( \sum_{j \in \mathcal{N}_i} \frac{\alpha_{ij} \log \alpha_{ij}}{\log |\mathcal{N}_i|} \right)$$

where $\mathcal{V}$ is the set of nodes of the graph and $\mathcal{N}_i$ is the neighbourhood set of node $i$.

Table 14: Average normalised entropy of attention scores for GNN and HGNN models on highest agent density scenarios across different maps. Results are averaged over 128 instances, and their 95% confidence intervals are reported.

| Method | Sparse Maze | Empty Room | Dense Maze | Dense Room | Dense Warehouse |
|---|---|---|---|---|---|
| GNN | $0.659 \pm 0.000$ | $0.662 \pm 0.000$ | $0.715 \pm 0.001$ | $0.733 \pm 0.001$ | $0.651 \pm 0.000$ |
| HGNN | $0.514 \pm 0.000$ | $0.502 \pm 0.000$ | $0.565 \pm 0.001$ | $0.592 \pm 0.001$ | $0.523 \pm 0.000$ |

Table 14 shows the average normalised entropy of attention scores for the GNN and HGNN models. We see that the HGNN model consistently has a significantly lower average normalised entropy of attention scores across all maps, indicating that GNNs suffer from attention dilution, while HGNNs can mitigate them.

## I    HYPERPARAMETER SENSITIVITY

We trained our HMAGAT ($k$-means) model with $k = 10\%$ of the number of vertices in the MAPF grid, $|V|$, and with communication radius $R^{\text{comm}} = 7$. In the following tables, we present the performance of our HMAGAT model with varying values of $k$ and $R^{\text{comm}}$ during test time.

Table 15 and Table 16 show the performance of HMAGAT ($k$-means) with varying values of $k$ and $R^{\text{comm}}$, respectively. We see that the performance of HMAGAT ($k$-means) is relatively stable across different values of $k$ and $R^{\text{comm}}$. The success rates vary slightly, but this variation might also be due

Table 15: Hyperparameter sensitivity analysis of HMAGAT ($k$-means), with varying values of $k$ on the highest agent density instances. Results are averaged over 128 instances. 95% confidence intervals are reported for Rel. SoC.

| **Metric** | $k$ | Sparse Maze | Empty Room | Dense Maze | Dense Room | Dense Warehouse |
|---|---|---|---|---|---|---|
| Success Rate | 5% | 90.6% | 100.0% | 62.5% | 60.2% | 82.0% |
| | 10% | 93.0% | 100.0% | 73.4% | 60.9% | 89.1% |
| | 20% | 91.4% | 100.0% | 68.8% | 61.7% | 93.0% |
| Rel. SoC | 5% | $1.45 \pm 0.03$ | $1.36 \pm 0.01$ | $1.31 \pm 0.07$ | $1.64 \pm 0.05$ | $1.04 \pm 0.05$ |
| | 10% | $1.47 \pm 0.04$ | $1.34 \pm 0.01$ | $1.27 \pm 0.06$ | $1.63 \pm 0.05$ | $1.02 \pm 0.05$ |
| | 20% | $1.45 \pm 0.03$ | $1.34 \pm 0.01$ | $1.26 \pm 0.06$ | $1.62 \pm 0.05$ | $1.02 \pm 0.05$ |

Table 16: Hyperparameter sensitivity analysis of HMAGAT ($k$-means), with varying values of $R^{\text{comm}}$ on the highest agent density instances. Results are averaged over 128 instances. 95% confidence intervals are reported for Rel. SoC.

| **Metric** | $R^{\text{comm}}$ | Sparse Maze | Empty Room | Dense Maze | Dense Room | Dense Warehouse |
|---|---|---|---|---|---|---|
| Success Rate | 5 | 93.8% | 100.0% | 73.4% | 60.2% | 89.1% |
| | 7 | 93.0% | 100.0% | 73.4% | 60.9% | 89.1% |
| | 9 | 93.0% | 100.0% | 73.4% | 66.4% | 90.6% |
| Rel. SoC | 5 | $1.46 \pm 0.03$ | $1.35 \pm 0.02$ | $1.28 \pm 0.06$ | $1.63 \pm 0.04$ | $1.01 \pm 0.05$ |
| | 7 | $1.47 \pm 0.04$ | $1.34 \pm 0.01$ | $1.27 \pm 0.06$ | $1.63 \pm 0.05$ | $1.02 \pm 0.05$ |
| | 9 | $1.46 \pm 0.03$ | $1.34 \pm 0.01$ | $1.25 \pm 0.06$ | $1.62 \pm 0.05$ | $1.03 \pm 0.05$ |

to randomness introduced by the sampling-based nature of the models. The relative SoC values remain robust across different hyperparameter settings, indicating that the solution quality of HMAGAT ($k$-means) is not significantly affected by these hyperparameters.

