# OpenReview forum: "Pairwise is Not Enough: Hypergraph Neural Networks for Multi-Agent Pathfinding"
_ICLR.cc/2026/Conference — ICLR 2026 Poster_

### Official Review · Reviewer_YpAe · 2025-10-30

**Soundness:** 2
**Presentation:** 3
**Contribution:** 2
**Rating:** 4
**Confidence:** 3

**Summary:**

This paper proposes HMAGAT, a hypergraph neural network-based approach for Multi-Agent Path Finding (MAPF). The authors argue that existing GNN-based methods are fundamentally limited by pairwise message passing, which leads to attention dilution in dense scenarios and fails to capture group interactions. HMAGAT leverages directed hypergraphs with attention mechanisms to explicitly model higher-order agent interactions. The method achieves impressive empirical results, matching or outperforming MAPF-GPT (85M parameters) while using only 1M parameters and training on 100× less data. The paper includes comprehensive experiments across 10 map types, detailed ablation studies, and insightful analyses using attention distributions and Shapley values.

**Strengths:**

The core motivation is strong and well-articulated. The observation that pairwise interactions are insufficient for highly-coupled multi-agent coordination problems is intuitive and important. Bringing hypergraph representations to MAPF is a natural and valuable direction—recent work has shown promise for hypergraphs in other multi-agent settings (e.g., trajectory prediction, formation control), and this paper extends that thinking to a challenging planning domain. The research question "Can hypergraphs scale beyond simple group settings to capture complex, highly-coupled multi-agent tasks?" is well-posed and addresses a real gap in the literature.

The attention dilution problem is clearly articulated and convincingly demonstrated. Figure 6 shows that GNN models spread attention across many agents in the middle range, preventing high focus on truly critical interactions, while HGNN maintains better differentiation. The hand-crafted Scenario 1 illustrates this effectively—as irrelevant agents are added to less important regions, GNN's attention to important agents becomes increasingly variable, while HGNN remains stable. This is a genuine problem that matters in practice.

The Shapley value analysis in Scenario 2 provides valuable interpretability. The results effectively demonstrate that the model captures group coordination patterns—showing substantially higher influence for agents involved in coupled interactions compared to isolated ones, despite symmetric spatial positions. This kind of mechanistic understanding is useful and relatively uncommon in multi-agent learning papers.

The experimental evaluation is comprehensive and well-designed. Testing across 10 diverse map types with varying densities provides solid evidence. The Dense Warehouse results are particularly striking, showing dramatic improvements in success rates at high agent densities where baseline methods struggle significantly. The ablation study properly isolates contributions, and the progression shows consistent improvement. The paper is clearly written with good visualizations, and the promised code release aids reproducibility.

The parameter efficiency and inference speed improvements are practically significant. These gains matter for real-world deployment in warehouse automation or traffic management scenarios.

**Weaknesses:**

My primary concern is that while the motivation is sound, the execution lacks depth. The architecture in Equations (1)-(4) adopts standard hypergraph attention formulations without meaningful adaptation to MAPF's specific structure. I appreciate that you're bringing hypergraphs to this domain, but the technical contribution feels more like careful engineering than methodological advancement. For instance, you could exploit MAPF-specific structure—agents have goals, there are collision constraints, paths have temporal dynamics—but the hypergraph construction treats this as a generic spatial proximity problem. The Lloyd/k-means/shortest-distance strategies are standard clustering approaches applied fairly directly. Given that hypergraphs have been explored in related multi-agent domains, I'd expect deeper innovation in how the representation is tailored to pathfinding constraints and objectives.

The training data disparity undermines the paper's central claim. You train on 21K instances while MAPF-GPT uses 3.75M—a 100× difference. The conclusion that "appropriate inductive biases are often more critical than training data size or sheer parameter count" is stated boldly but not properly validated. You're comparing two confounded variables simultaneously (architecture AND data scale), making it impossible to isolate the effect. It's entirely plausible that MAPF-GPT is simply undertrained relative to its capacity. I recognize that generating millions of expert trajectories is expensive, but at minimum, you should show MAPF-GPT trained on your 21K instances to establish whether the gap persists under equal data conditions. This is a fundamental experimental control that's missing, and it weakens confidence in the main message.

The theoretical foundation is thin. You provide no formal analysis of when or why hypergraphs should outperform GNNs beyond intuition and two constructed scenarios. What types of coordination patterns can HGNNs represent that GNNs cannot? Are there expressiveness results or approximation guarantees? The attention dilution observation is insightful but empirical—showing it occurs systematically across many real benchmark instances (not just hand-crafted examples) would be more convincing. The paper would benefit substantially from even informal theoretical characterization of the representational differences.

The hypergraph generation lacks principled justification. Why these specific strategies? The shortest-distance method shows mixed results across different maps and has higher computational overhead, yet there's no clear guidance on when each strategy is appropriate. The "soft boundary" operation (discarding half the colors) appears to be a critical tuning choice but receives minimal discussion—what happens with different percentages? More fundamentally, there's no framework for thinking systematically about hypergraph design for MAPF. Should edges capture potential conflicts? Goal proximity? Historical collision patterns? The approach feels somewhat ad-hoc rather than principle-driven.

The failure analysis is missing. Even in scenarios where HMAGAT performs best, failure rates remain. What are these failure modes? Are they timeouts, deadlocks, or inefficient paths? Understanding when hypergraphs don't help would provide important insight into the method's limitations and scope.

**Questions:**

1. The 100× training data difference between HMAGAT (21K) and MAPF-GPT (3.75M) is my biggest concern. Can you train MAPF-GPT on your 21K instances and report results? This would isolate whether your gains come from architectural improvements or simply from the baseline being undertrained.

2. Can you provide some theoretical analysis—even informal—about what makes hypergraphs better suited for MAPF?

3. What are the typical failure modes when HMAGAT doesn't succeed? Understanding when hypergraphs fail would help clarify their scope of applicability.

4. Can you demonstrate that attention dilution happens systematically in real benchmark runs, not just the two hand-crafted scenarios? Maybe show attention statistics across many Dense Warehouse episodes?

5. How much wall-clock time does training take for HMAGAT compared to MAGAT? What's the computational overhead of hypergraph construction?

6. What guidance would you give practitioners for choosing between Lloyd, k-means, and shortest-distance strategies? When does each work best?

7. How sensitive is performance to hyperparameters like k (number of colors), R^comm, and R^obs?

The core idea here is interesting and the paper is generally well-written. The motivation for using hypergraphs in dense coordination scenarios resonates with me, and the empirical results look promising. That said, the training data confound and lack of theoretical grounding  are issues that need addressing. I'm also curious about the failure analysis  and whether attention dilution is a systematic phenomenon  or mainly visible in constructed examples. If the rebuttal can provide satisfactory answers to these concerns, particularly the first two, I'd be happy to reconsider my score.

---

> ### Author Response · Authors · 2025-11-21
>
> Dear Reviewer,
>
> Thank you for your insightful comments. Below, we provide our responses. We are still working on additional experiments and analyses, and will provide them later.
>
> **Q1 (data size diff):** To answer this question precisely, we are currently training the MAPF-GPT models on our dataset. This training is slow, and we hope to post results in the coming days–thank you for your patience!
>
> Although the training is not done, we are certain that this model would just be a weaker version of the original MAPF-GPT model. This is because of lack of diversity of data against model size. The original model training involves more number of iterations than our training, and this means that our training won’t help with overcoming the undertraining of the model (assuming it was originally undertrained).
>
> **Q2 (theoretical analysis):** MAPF is a NP-hard problem, involving complex highly coupled multi-agent dynamics. There does not exist a known ground truth representation for the inter-agent interactions. Graphs (pairwise) and hypergraphs (group) are two ways of modelling these interactions. In Fig. 1(c), we show how a purely pairwise perspective of the interactions leads to sub-optimal solutions compared to a group interaction model. These indicate that hypergraphs should be a better representation for MAPF than graphs, which is further supported by the empirical evidence provided in Sec. 4.
>
> We will provide further theoretical analysis on attention dilution in the upcoming days.
>
> **Q3 (failure modes):** Below, we provide an analysis of the failure cases of HMAGAT on the Dense Room maps, the map class with lowest success rate for HMAGAT.
>
> First, we note that even in scenarios where HMAGAT fails in the Dense Room maps, it achieves an average of 97.3% partial success rate (percentage of agents that reach their goals at the end of task). Of the 2.7% agents that failed to reach their goals at the final timestep, 90% reached their goal sometime during the programme execution, but later had to move. The table below reports the failure mode for these 2.7% of agents that fail to reach their goals.
>
> | Failure Mode | Percentage |
> |--------------|------------|
> | Deadlock | 3.4% |
> | Livelock | 46.4% |
> | Hit max timestep limit | 50.2% |
>
> Here, we say an agent was deadlocked if, for the last 5 actions, it stayed at the same location. An agent was said to be livelocked, if it only oscillated between locations at least 3 times before hitting the max timestep limit. Rest of the agents were assumed to have failed due to simply hitting the max timestep limit. Interestingly, all the livelocks consist of oscillation between only two locations. Given this, it should be possible to improve the success rate of HMAGAT, using some straightforward deadlock/livelock detection mechanism, escaping the deadlock/livelock by increasing the sampling temperature or following a greedy path (as done by https://arxiv.org/abs/2510.17382), whenever a deadlock/livelock is detected.
>
> **Q4 (attention dilution):**  We will answer this in the upcoming days.
>
> **Q5 (hypergraph computational overhead):** HMAGAT training has a 6% training overhead compared to its graph counterpart. With regards to the hypergraph construction time, given a moderate graph size, this cost is not prohibitive with respect to the model inference. E.g., on the Dense Warehouse map with 128 agents, HMAGAT takes on average 19s to solve, out of which 0.04s are spent on the colouring of the map. 0.005s (~8% of inference time) are spent on constructing the hypergraphs from the colouring at each time step. In practice, this can be highly optimized using C++. We will clarify this in the updated version of the paper.
>
> **Q6 (hypergraph strategies):** Although the use of hypergraphs has been explored in other multi-agent domains, in these domains, either the effective group interaction has already been identified or the interactions are simple and easy to predict. On the other hand, in MAPF, the effective group interaction is unknown and involves complex joint coordination. Therefore, we provided three reasonable, principled examples (Lloyd, k-means and shortest distance-based) and empirically compared them. We acknowledge that there is a design choice in hypergraph generation. Based on the results, we generally recommend using the k-means method, given its empirical performance in terms of success rate, runtime, and solution cost across diverse scenarios. A future line of research could be the automatic generation of hypergraphs, or more advanced hypergraph generation specific to MAPF as the reviewer suggested, but these are beyond the scope of this study.
>
> **Q7 (hyperparameter sensitivity):**  We will answer this in the upcoming days.

---

> ### Author Response · Authors · 2025-11-26
> **Official Comment by Authors answering Remaining Questions [1/3]**
>
> Dear Reviewer,
>
> Thank you for your patience. Below, we provide our responses to the remaining questions.
>
> **Q1 (data size diff -- continued):** We were able to finish training MAPF-GPT (2M) on our dataset with 21K instances. However, we find that it does not train well, obtaining a mere 60% training and validation accuracy after the training is done (for comparison, HMAGAT obtains 95+% accuracy). The model might be hyperparameter sensitive. We have used the same hyperparameters as reported in the appendix of the MAPF-GPT paper. We also experimented with using the learning rate we used for HMAGAT, but we still obtained similar results. A note: we do not prune our dataset to discard 80% of the wait-at-target actions, as done in the MAPF-GPT paper.
>
> Thus, likely due to sensitivity to hyperparameters or dataset quality and/or quantity, MAPF-GPT (2M) was unable to train on our dataset. However, as we stated earlier, we do not believe MAPF-GPT would benefit from training on our dataset, as it would just be further undertrained, resulting in a weaker model.
>
> **Q2 (attention dilution theoretical analysis):** We provide an informal analysis, demonstrating that GNNs face attention dilution, which HGNNs are capable of mitigating. We first start with demonstrating that GNNs suffer from attention dilution:
>
> ### GNNs suffer from attention dilution
>
> Consider a graph $\mathcal{G} = (\mathcal{V}, \mathcal{E})$ and a node $i \in \mathcal{V}$ with neighbourhood $\mathcal{N}\_i$, where we can partition $\mathcal{N}\_i$ into two subsets:
> * $\mathcal{N}\_i^{\text{noise}}$: nodes in a similar locality that do not aid in predicting $i$'s label.
> * $\mathcal{N}\_i^{\text{rest}}$: the remaining nodes in the neighbourhood.
>
> Let $a\_{ij}$ denote the pre-softmax attention score for node $j \in \mathcal{N}\_i$ with respect to node $i$, and let the normalized attention scores be given by $\alpha\_{ij} = \frac{\text{exp}(a\_{ij})}{\sum\_{k \in \mathcal{N}\_i} \text{exp}(a\_{ik})}$.
>
> To demonstrate attention dilution, we construct a new graph $\mathcal{G}^\prime = (\mathcal{V}^\prime, \mathcal{E}^\prime)$ by adding more noisy nodes to graph $\mathcal{G}$. Specifically, we define the neighbourhood of node $i$ in the new graph as $\mathcal{N}\_i^\prime = \mathcal{N}\_i \cup A^\prime$, where nodes in $A^\prime$ are in a similar locality as $\mathcal{N}\_i^{\text{noise}}$. The other nodes retain their original features. Let $\mathbf{x}\_j^\prime$ and $\omega\_{ij}^\prime$ represent the node embeddings and edge vectors in the new graph, with $a\_{ij}^\prime$ and $\alpha\_{ij}^\prime$ denoting the pre-softmax and normalized attention scores, respectively.
>
> Since, $\mathbf{x}\_j^\prime = \mathbf{x}\_j$ and $\omega\_{ij}^\prime = \omega\_{ij}$ for all $j \in \mathcal{N}\_i$, we have $a\_{ij}^\prime = a\_{ij}$ for all $j \in \mathcal{N}\_i$.
>
> Thus, for all nodes $j \in \mathcal{N}\_i^{\text{rest}}$, we have:
> $$\alpha\_{ij}^\prime = \frac{\text{exp}(a\_{ij}^\prime)}{\sum\_{k \in \mathcal{N}\_i^\prime} \text{exp}(a\_{ik}^\prime)} = \frac{\text{exp}(a\_{ij})}{\sum\_{k \in \mathcal{N}\_i} \text{exp}(a\_{ik}) + \sum\_{k \in A^\prime} \text{exp}(a\_{ik}^\prime)} < \alpha\_{ij}$$
>
> This shows that the attention scores for nodes in $\mathcal{N}\_i^{\text{rest}}$ have decreased due to the addition of noisy nodes in $A^\prime$, demonstrating attention dilution.

---

> ### Author Response · Authors · 2025-11-26
> **Official Comment by Authors answering Remaining Questions [2/3]**
>
> ### HGNNs can mitigate attention dilution
>
> Now, we demonstrate that HGNNs can mitigate attention dilution:
>
> Consider a hypergraph $\mathcal{H} = (\mathcal{V}, \mathcal{E}\_H)$ corresponding to the same instance represented by the graph $\mathcal{G}$. Again, let node $i \in \mathcal{V}$ have a node neighbourhood that can be partitioned into two subsets -- $\mathcal{N}\_i^{\text{noise}}$ and $\mathcal{N}\_i^{\text{rest}}$, with nodes in $\mathcal{N}\_i^{\text{noise}}$ being in a similar locality. Assume an appropriate hypergraph structure such that nodes in $\mathcal{N}\_i^{\text{noise}}$ belong to the tail of a single hyperedge $e\_\sigma \in \mathcal{E}\_H$, while nodes in $\mathcal{N}\_i^{\text{rest}}$ belong to the tails of other hyperedges, $\mathcal{E}\_{\text{rest}} \subset \mathcal{E}\_H$, with $i$ being in the head of all these hyperedges. Let $\alpha\_{ej}$ be the attention score for node $j$ with respect to hyperedge $e$, and let $a\_{ie}$ and $\alpha\_{ie}$ be the pre-softmax and normalized attention scores for hyperedge $e$ with respect to node $i$, respectively. Here we have:
> $$\alpha\_{ie} = \frac{\text{exp}(a\_{ij})}{\sum\_{k \in \{e\_\sigma\} \cup \mathcal{E}\_{\text{rest}}} \text{exp}(a\_{ik})}$$
>
> Also, let the hyperedge embedding for an hyperedge $e$ be $\mathbf{h}\_{e}$.
>
> Now, we construct the new hypergraph $\mathcal{H}^\prime = (\mathcal{V}^\prime, \mathcal{E}\_H^\prime)$, corresponding to the same instance represented by the graph $\mathcal{G}^\prime$. Here we have $\mathcal{V}^\prime = \mathcal{V} \cup A^\prime$, with nodes in $A^\prime$ having similar locality to nodes in $\mathcal{N}\_i^{\text{noise}}$. Since they share a locality, we can assume that nodes in $\mathcal{N}\_i^{\text{noise}} \cup A^\prime$ belong to the tail of the same hyperedge $e\_\sigma^\prime \in \mathcal{E}\_H^\prime$, while nodes in $\mathcal{N}\_i^{\text{rest}}$ belong to the tails of other hyperedges, $\mathcal{E}\_{\text{rest}}$, with $i$ being in the head of all the hyperedges, similar to the original hypergraph.
>
> For all nodes $j \in \mathcal{N}\_i^{\text{rest}}$, we have:
> $$\alpha\_{ej}^\prime = \alpha\_{ej}$$
> This is because the hyperedges in $\mathcal{E}\_{\text{rest}}$ have the same tail nodes as in the original hypergraph, leading to the same attention scores. This means the hyperedge embeddings for hyperedges in $\mathcal{E}\_{\text{rest}}$ remain unchanged, i.e. $\mathbf{h}\_e^\prime = \mathbf{h}\_e$ for all $e \in \mathcal{E}\_{\text{rest}}$. This also means that the pre-softmax attention scores for hyperedges in $\mathcal{E}\_{\text{rest}}$ remain unchanged, i.e. $a\_{ie}^\prime = a\_{ie}$ for all $e \in \mathcal{E}\_{\text{rest}}$.
>
> With regards to hyperedge $e\_\sigma^\prime$: The original hyperedge $e\_\sigma$ consisted of nodes not helpful with $i$'s label prediction. Since the hyperedge embeddings are computed as a weighted combination of the node embeddings, simply adding more nodes to the tail of the hyperedge does not increase the magnitude or relevance of the hyperedge embedding.
> Since the new nodes, $A^\prime$, are also not helpful for predicting node $i$'s label, they will not increase the hyperedge's attention score. Thus, we can assume that the pre-softmax attention score for hyperedge $e\_\sigma^\prime$ remains unchanged, i.e. $a\_{i e\_\sigma^\prime}^\prime = a\_{i e\_\sigma}$.
>
> Thus, for all nodes $j \in \mathcal{N}\_i^{\text{rest}}$, we have:
> $$\alpha\_{ij}^\prime = \frac{\text{exp}(a\_{ij}^\prime)}{\sum\_{k \in \{e\_\sigma^\prime\} \cup \mathcal{E}\_{\text{rest}}} \text{exp}(a\_{ik}^\prime)} = \frac{\text{exp}(a\_{ij})}{\sum\_{k \in \{e\_\sigma\} \cup \mathcal{E}\_{\text{rest}}} \text{exp}(a\_{ik})} = \alpha\_{ij}$$
>
> This shows that the attention scores for nodes in $\mathcal{N}\_i^{\text{rest}}$ remain unchanged despite the addition of noisy nodes in $A^\prime$, demonstrating that HGNNs can mitigate attention dilution. Thus, unlike in GNNs, introducing additional noisy nodes does not dilute the attention assigned to informative nodes.

---

> ### Author Response · Authors · 2025-11-26
> **Official Comment by Authors answering Remaining Questions [3/3]**
>
> **Q4 (attention dilution):** As a relative measure of the attention dilution, we calculate the average normalized entropy of the attention scores, averaged across all execution steps and 128 instances, using the formula below:
>
> $$H = -\sum\_{i \in \mathcal{V}} \frac{1}{\lvert \mathcal{V} \rvert} \left( \sum\_{j \in \mathcal{N}\_i}\frac{\alpha\_{ij} \log\alpha\_{ij}}{\log \lvert \mathcal{N}\_i \rvert} \right)$$
>
> Here $\mathcal{V}$ is the set of nodes of the graph and $\mathcal{N}_i$ is the neighbourhood set of node $i$.
>
> A higher value of entropy, $H$, correlates with a higher dilution of attention. Below we show the average normalized entropy values (the lower the better) for HGNN (stripped down HMAGAT) and GNN (MAGAT) along with their 95% confidence intervals, calculated on the highest agent density instances:
>
> | Method | Sparse Maze | Empty Room | Dense Maze | Dense Room | Dense Warehouse |
> |--------|-------------|------------|------------|------------|-----------------|
> | GNN | $0.659 \pm 0.000$ | $0.662 \pm 0.000$ | $0.715 \pm 0.001$ | $0.733 \pm 0.001$ | $0.651 \pm 0.000$ |
> | HGNN | $0.514 \pm 0.000$ | $0.502 \pm 0.000$ | $0.565 \pm 0.001$ | $0.592 \pm 0.001$ | $0.523 \pm 0.000$ |
>
> We see that HGNN consistently obtains lower entropy than GNN, showing that attention dilution happens systematically across the datasets.
>
> **Q7 (hyperparameter sensitivity):** We trained our k-means-based HMAGAT model with $k = 10\%$ of the number of vertices in the MAPF grid, $\lvert V \rvert$, and with communication radius $R^{\text{comm}} = 7$. In the following tables, we present the performance of our HMAGAT model with varying values of $k$ and $R^{\text{comm}}$ during test time.
>
> The below table shows the success rates of our HMAGAT model with varying values of $k$:
>
> | Algorithm | Sparse Maze | Empty Room | Dense Maze | Dense Room | Dense Warehouse |
> |-----------|-------------|------------|------------|------------|-----------------|
> | k = 5% | 90.6\% | 100.0\% | 62.5\% | 60.2\% | 82.0\% |
> | k = 10% | 93.0\% | 100.0\% | 73.4\% | 60.9\% | 89.1\% |
> | k = 20% | 91.4\% | 100.0\% | 68.8\% | 61.7\% | 93.0\% |
>
> We also present the Rel. SoC for these:
>
> | Algorithm | Sparse Maze | Empty Room | Dense Maze | Dense Room | Dense Warehouse |
> |-----------|-------------|------------|------------|------------|-----------------|
> | k = 5% | $1.45 \pm 0.03$| $1.36 \pm 0.01$| $1.31 \pm 0.07$| $1.64 \pm 0.05$| $1.04 \pm 0.05$|
> | k = 10% | $1.47 \pm 0.04$| $1.34 \pm 0.01$| $1.27 \pm 0.06$| $1.63 \pm 0.05$| $1.02 \pm 0.05$|
> | k = 20% | $1.45 \pm 0.03$| $1.34 \pm 0.01$| $1.26 \pm 0.06$| $1.62 \pm 0.05$| $1.02 \pm 0.05$|
>
> These results indicate that our HMAGAT model is relatively robust to the choice of $k$.
>
> The below table shows the success rates of our HMAGAT model with varying values of $R^{\text{comm}}$:
>
> | Algorithm | Sparse Maze | Empty Room | Dense Maze | Dense Room | Dense Warehouse |
> |-----------|-------------|------------|------------|------------|-----------------|
> | $R^{\text{comm}} = 5$ | 93.8\% | 100.0\% | 73.4\% | 60.2\% | 89.1\% |
> | $R^{\text{comm}} = 7$ | 93.0\% | 100.0\% | 73.4\% | 60.9\% | 89.1\% |
> | $R^{\text{comm}} = 9$ | 93.0\% | 100.0\% | 73.4\% | 66.4\% | 90.6\% |
>
> We also present the Rel. SoC for these:
>
> | Algorithm | Sparse Maze | Empty Room | Dense Maze | Dense Room | Dense Warehouse |
> |-----------|-------------|------------|------------|------------|-----------------|
> | $R^{\text{comm}} = 5$ | $1.46 \pm 0.03$| $1.35 \pm 0.02$| $1.28 \pm 0.06$| $1.63 \pm 0.04$| $1.01 \pm 0.05$|
> | $R^{\text{comm}} = 7$ | $1.47 \pm 0.04$| $1.34 \pm 0.01$| $1.27 \pm 0.06$| $1.63 \pm 0.05$| $1.02 \pm 0.05$|
> | $R^{\text{comm}} = 9$ | $1.46 \pm 0.03$| $1.34 \pm 0.01$| $1.25 \pm 0.06$| $1.62 \pm 0.05$| $1.03 \pm 0.05$|
>
> These results also indicate that our HMAGAT model is relatively robust to the choice of $R^{\text{comm}}$.
>
> The variation of hyperparameter $R^{\text{obs}}$ would require retraining the model. We have used the same value as used by MAPF-GPT.

---

### Official Review · Reviewer_KnuG · 2025-10-31

**Soundness:** 2
**Presentation:** 3
**Contribution:** 2
**Rating:** 4
**Confidence:** 4

**Summary:**

The paper introduces HMAGAT, a hypergraph-based extension of the MAGAT framework for multi-agent pathfinding. The core idea is to capture higher-order agent interactions beyond pairwise relationships by representing agent dependencies through hyperedges. The method integrates several enhancements, including observation encoding improvements, post-training quality optimization, and temperature-based sampling during inference. The authors evaluate HMAGAT across multiple benchmark MAPF environments, comparing it to state-of-the-art learning-based methods.

**Strengths:**

- The paper is generally well-written and clearly structured. The graphical illustrations are clear and informativel.
- The experimental evaluation includes a solid and well-chosen set of test maps.
- HMAGAT demonstrates strong empirical performance across a diverse set of benchmarks, showing both scalability and robustness.

**Weaknesses:**

W1: The motivational example in Figure 1(c) is unclear, and I do not fully understand what prevents models such as MAPF-GPT from learning an optimal policy when expert data for this case are provided. It would be helpful if the authors clarified what exactly is meant by pairwise interactions in MAPF and why these interactions would limit a learnable model from reproducing the expert’s optimal SoC solution. A formal definition or additional explanation would make the example more convincing.

W2: Based on the ablation results in Figure 5, the final performance appears to stem from multiple components, including Hypergraphs, OE Quality Improvement, Post-Train Quality Improvement, and Temperature Sampling. As a result, the initial motivation regarding pairwise interactions seems less supported, and the impact of the hypergraph component does not appear dominant. This also suggests that the paper’s focus on hypergraphs for highly-coupled multi-agent problems is somewhat overshadowed by the combination of techniques used to outperform current learnable MAPF approaches.

W3: The proposed approach builds on several prior methods and involves a multi-stage, complex pipeline, including the MAGAT base, the addition of Hypergraphs and a mechanism to prepare observations, offline training on expert data, subsequent online fine-tuning, and a separate RL-trained model for temperature sampling. While this combination achieves strong performance, it raises concerns about the overall simplicity and generality of the approach. Relying on heavily staged, task-specific designs may limit scalability and the ability to leverage general computation, suggesting that more unified or end-to-end approaches could be more broadly effective.

W4:  I’m not entirely convinced that collision shielding should be applied to all approaches (as recommended by Veerapaneni et al., 2025). This suggestion is not theoretically motivated, and based on decision-making theory, it effectively changes the policy. We can consider two policies: $\pi$, the default one, and $sh(\pi)$, the policy with shielding. When performing imitation learning from an expert, applying shielding introduces a distributional shift.

For MAPF-GPT-85M, this works well because the model is trained on expert data where no collisions occur, so shielding helps, even with the shift between PIBT and LaCAM. However, for MAPF-GPT-DDG-2M, this is not the case, since the model was fine-tuned and the model performance was influenced by its own behavior.

Formally, since the policy determines the trajectory of states, applying shielding changes the state distribution: $s_{t+1} \sim P(\cdot \mid s_t, a_t), \quad a_t \sim \pi(a_t \mid s_t)$ versus $s'_{t+1} \sim P(\cdot \mid s'_t, a'_t), \quad a'_t \sim sh(\pi)(a'_t \mid s'_t),$

which implies that $p(s_t) \neq p(s'_t)$ and consequently $p(o_t) \neq p(o'_t)$, where $o_t$ denotes agent' observation at time $t$. Thus, shielding induces a distinct observation distribution, on which the model was not trained.

The proper approach, in my view, is to fine-tune MAPF-GPT-DDG in an environment that includes shielding to eliminate this inconsistency.

W5: Following up on the previous weakness, I believe it is not entirely fair to apply modifications to existing approaches without conducting a proper ablation study. It is not clear how shielding influences the algorithms to which it was applied. While the authors provide some ablation results in Table 5 of the appendix, a more thorough analysis would involve testing all approaches both with and without shielding, and reporting the complete set of metrics (including success rate, SoC, and runtime).

W6: It seems to be an overstatement in the experimental results that HMAGAT consistently outperforms larger models, except for MAPF-GPT-85M. For example, MAPF-GPT-DDG (2M) shows better performance in instances with a large number of agents for the Sparse Maze and Dense Room scenarios, and comparable performance in the Empty Room and Dense Maze environments. Moreover, despite achieving better or similar success rates, MAPF-GPT-DDG exhibits lower performance in terms of SoC, which appears counterintuitive. It would be helpful if the authors investigated this further and provided an explanation of how these plots were generated.

Despite the major concerns raised, I remain open to discussion and willing to adjust my score depending on how the identified weaknesses are addressed.

**Questions:**

Q1: How does adding PIBT shielding affect the model’s runtime performance?

Q2: The runtime spikes observed seem unusual for the MAPF-GPT family (especially on _ost003d_). Did the authors investigate this behavior, and how exactly was runtime measured or computed?

Q3: Were the authors willing to provide the raw testing data or the code  to run a full evaluation (script to run all testing instances) of the proposed approach?

Q4: How much GPU time is required to reproduce the testing results?

---

> ### Author Response · Authors · 2025-11-21
> **Official Comment by Authors [1/2]**
>
> Dear Reviewer,
>
> Thank you for your insightful comments. Below, we provide our responses. We are still working on additional experiments and analyses, and will provide them later.
>
> **Q1, W4 & W5 (PIBT collision shielding; CS):** We first note that this distribution shift is not specific to PIBT. All learned policies for MAPF alone cannot guarantee collision-freeness; therefore, their outputs are post-processed by the so-called naive CS, which is provided as part of the POGEMA benchmark. PIBT is simply an improved version of this naive approach. Moreover, we empirically confirmed that all policies, including MAPF-GPT-DDG (2M), achieved significantly better performance when combined with PIBT, while incurring negligible overhead compared to neural inference. This observation is consistent with several studies:  https://arxiv.org/abs/2403.20300 https://arxiv.org/abs/2510.17382 . Given these results, we see little incentive to use the vanilla/naive version.
>
> The following data, showing the success rate of MAPF-GPT models w/ and w/o PIBT collision shielding on the highest agent density evaluations, supports the above.
>
> | Algorithm | Sparse Maze | Empty Room | Dense Maze | Dense Room | Dense Warehouse |
> |-----------|-------------|------------|------------|------------|-----------------|
> | MAPF-GPT (2M) | 77.3\% | 100.0\% | 39.8\% | 53.9\% | 0.0\% |
> | MAPF-GPT (2M) w/o CS-PIBT | 0.0\% | 0.0\% | 0.0\% | 0.0\% | 0.0\% |
> | MAPF-GPT (6M) | 93.0\% | 100.0\% | 58.6\% | 74.2\% | 3.9\% |
> | MAPF-GPT (6M) w/o CS-PIBT | 0.0\% | 0.0\% | 0.0\% | 0.0\% | 0.0\% |
> | MAPF-GPT-DDG (2M) | 99.2\% | 100.0\% | 73.4\% | 86.7\% | 10.9\% |
> | MAPF-GPT-DDG (2M) w/o CS-PIBT | 34.4\% | 46.1\% | 10.2\% | 7.0\% | 0.0\% |
>
> We also show the Rel. SoC of these models below, also supporting the usefulness and importance of PIBT collision shielding:
>
> | Algorithm | Sparse Maze | Empty Room | Dense Maze | Dense Room | Dense Warehouse |
> |-----------|-------------|------------|------------|------------|-----------------|
> | MAPF-GPT (2M) | 2.146 | 1.643 | 3.167 | 3.092 | 2.390 |
> | MAPF-GPT (2M) w/o CS-PIBT | 8.043 | 6.834 | 6.791 | 7.960 | 2.390 |
> | MAPF-GPT (6M) | 1.770 | 1.486 | 2.756 | 2.541 | 2.271 |
> | MAPF-GPT (6M) w/o CS-PIBT | 7.528 | 6.437 | 6.507 | 7.591 | 2.396 |
> | MAPF-GPT-DDG (2M) | 1.788 | 1.560 | 2.666 | 2.196 | 2.184 |
> | MAPF-GPT-DDG (2M) w/o CS-PIBT | 5.387 | 4.484 | 5.511 | 6.383 | 2.377 |
>
> Lastly, the table below details the per-map runtime (in seconds) for the MAPF-GPT models w/ and w/o PIBT collision shielding.
>
> | Algorithm | Sparse Maze | Empty Room | Dense Maze | Dense Room | Dense Warehouse |
> |-----------|-------------|------------|------------|------------|-----------------|
> | MAPF-GPT (2M) | 10.5 | 5.7 | 20.9 | 21.7 | 23.6 |
> | MAPF-GPT (2M) w/o CS-PIBT | 13.4 | 14.1 | 20.3 | 20.8 | 24.9 |
> | MAPF-GPT (6M) | 19.1 | 10.7 | 34.2 | 31.7 | 37.8 |
> | MAPF-GPT (6M) w/o CS-PIBT | 22.4 | 23.5 | 35.3 | 38.5 | 41.4 |
> | MAPF-GPT-DDG (2M) | 6.8 | 4.9 | 19.8 | 15.5 | 23.6 |
> | MAPF-GPT-DDG (2M) w/o CS-PIBT | 11.9 | 12.6 | 19.7 | 20.4 | 23.3 |
>
> The models w/ PIBT collision shielding have, in general, a lower per-map runtime, due to the increased quality of solutions, leading to fewer inference steps.
>
> **Q2 (runtime spike of MAPF-GPT):** We used the publicly available code provided by the authors and measured the wall-clock time from start to finish with the help of GPU inference. The ost003d map is larger than the others and requires a greater number of inferences, resulting in longer runtime.
>
> **Q3 (code availability):** This has been included in the supplementary material, with instructions on how to reproduce our results. We plan to release it as open source software after publication.
>
> **Q4 (GPU time):** Reproducing the evaluation results in Fig. 4 will require around 150 GPU hours. A major portion of these will be spent on the ost003d evaluations, which requires around 90 GPU hours.
>
> **W1 (pairwise interaction’s limitation):** We will answer this in the upcoming days.

---

> ### Author Response · Authors · 2025-11-21
> **Official Comment by Authors [2/2]**
>
> **W2 & W3 (impact of hypergraphs, pipeline complexity):** We re-emphasise the goal of this paper: awareness of group interaction is important in MAPF, and hypergraphs provide a natural representation for this purpose. With this in mind, the paper empirically shows that (i) hypergraphs are more amenable than standard GNNs (Fig. 5, Table 1), and therefore (ii) HMAGAT can achieve state-of-the-art performance using training techniques commonly found in the literature, with superior training efficiency than prior work (Figs. 1 and 4). This structure does not undermine our original purpose; rather, we believe it effectively invites researchers who solely rely on pairwise modelling to reconsider their assumptions.
>
> We particularly note that the mechanism to prepare observations is standard, the quality improvement trainings are similar to MAPF-GPT-DDG, and the temperature sampling follows naturally from the need to address the underconfidence of GNNs and HGNNs. All of these can be easily adapted for imitation learning on other multi-agent tasks, and are not less general than an end-to-end approach, which would also require preparation of observations, advanced training schemes and hyperparameter and/or architecture tuning.
>
> **W6 (performance wrt. MAPF-GPT-DDG):** The results report was careful not to overclaim the strength of HMAGAT. We do not claim that HMAGAT outperforms MAPF-GPT-DDG (2M) *consistently*, because that would imply that HMAGAT consistently obtains both, higher quality solutions and better success rates, than MAPF-GPT-DDG (2M), which is not the case for the Dense Room scenarios. However, we claim that HMAGAT *consistently* obtains higher quality solutions than MAPF-GPT-DDG (2M), which is correct, even in the Dense Room scenarios, as seen by the Rel. SoC plots.
>
> Regarding Rel. SoC plots: We follow the strategy used by POGEMA, and, in turn, MAPF-GPT, where the costs for agents that fail to reach their goals are taken to be the episode length. For iterative solvers, this is a valid strategy to get a lower bound on the SoC if the episode length was not limited.
>
> SoC and success rates are not directly correlated metrics. A method can predict a successful but highly inefficient path, leading to a high success rate but poor SoC. On the other hand, a method whose solutions lead to most agents reaching their goal locations highly efficiently, but with some agents not reaching their goal locations, will have a low success rate but good SoC.
>
> On the Dense Room maps with 128 agents, we find that although HMAGAT achieves lower success rate than MAPF-GPT-DDG (2M), HMAGAT achieves a similar (in fact, slightly higher) partial success rate (average percentage of agents that reach their goal locations at the end of task) of 98.89% compared to MAPF-GPT-DDG (2M), which achieves a partial success rate of 98.86%. MAPF-GPT-DDG (2M) produces inefficient paths for the agents to reach their goals and so has a poorer SoC than HMAGAT.

---

> ### Author Response · Authors · 2025-11-26
> **Official Comment by Authors answering Remaining Questions**
>
> Dear Reviewer,
>
> Thank you for your patience. Below, we provide our responses to the remaining questions.
>
> **W1 (pairwise interaction’s limitation):** The example in Figure 1(c) illustrates that optimal MAPF solutions involve group interactions. We use it to provide intuition on why optimal MAPF solutions inherently involve group interactions, and thus, hypergraphs are a more natural representation for these problems. Below, we elaborate on this, providing more details on what we refer to as 'pairwise interactions' in the context of MAPF.
>
> MAPF solvers take as input information about each agent, and use this information to compute paths for all agents. For MAGAT/GNNs, each agent's information is represented using relevant node embeddings, while for MAPF-GPT/transformers, this information is represented using relevant token embeddings. An action for agent $i$ is predicted based on the information of agent $i$ and the information of other agents. When only pairwise interactions are considered, we say the action for agent $i$, $o_i$, is predicted as follows:
> $$o_i = \bigoplus_{j \in \mathcal{N}_i} \phi(\mathbf{x}_i, \mathbf{x}_j)$$
> where $\mathbf{x}_i$ is the information of agent $i$, $\phi$ is a function that models the interaction between agent $i$ and agent $j$ and $\bigoplus$ is an aggregation function for the interactions.
>
> Now, in the context of Figure 1(c), we show the pairwise interactions, $\phi$, in parts (iii-v), with respect to agent A. $\bigoplus$ can be taken to be any aggregation function; in this case, it could just be the intersection operation among the pairwise paths. This results in the predicted path for the agents as shown in part (ii). However, we see that this predicted path is not optimal, and the purely pairwise interactions fail to capture the group interaction that leads to the optimal solution shown in part (i).
>
> In the context of MAGAT and MAPF-GPT, $\phi(\mathbf{x}_i, \mathbf{x}_j)$ is the messages in the message passing framework and the value vector in the attention layer, respectively, while $\bigoplus$ is the attention-based weighted-mean aggregation in both cases.
>
> As noted in Section 3, although these mechanisms are only using pairwise interactions, they can still perform well on MAPF tasks, by learning to approximate group interactions. They can do so by: (i) using high-dimensional embeddings that can simply encode all the information about other agents, followed by a complex enough function to decode this information, or (ii) stacking multiple layers of pairwise interactions, which can allow for indirect capture of group interactions. Both MAGAT and MAPF-GPT use a combination of these two strategies. However, both of these strategies have their limitations.
>
> The first strategy relies on the model's ability to encode and decode all relevant information about other agents within high-dimensional embeddings. This may work well for small-scale problems, but as the number of agents increases, the amount of information that needs to be encoded grows significantly, making it increasingly difficult for the model to effectively capture all necessary interactions. The second strategy, stacking multiple layers of pairwise interactions, can help indirectly capture group interactions, but given a fixed number of layers, there is a limit to the complexity of group interactions that can be effectively modeled. As the number of agents increases, the depth required to capture all relevant group interactions may exceed practical limits.
>
> In contrast, hypergraph-based approaches can directly model the group interactions among agents, allowing for a more natural and efficient representation of the MAPF problem. Also, purely from an alignment perspective, hypergraph-based approaches align better with the underlying structure of interactions in optimal MAPF solutions.

---

### Official Review · Reviewer_VXsq · 2025-11-01

**Soundness:** 3
**Presentation:** 2
**Contribution:** 3
**Rating:** 6
**Confidence:** 3

**Summary:**

This paper proposes HMAGAT, a hypergraph-based imitation learning framework for Multi-Agent Path Finding (MAPF). It argues that pairwise interaction modeling, as commonly done via GNNs or transformers, is insufficient for capturing the inherently group-based dependencies among agents. By introducing directed hypergraph attention layers, HMAGAT captures higher-order interactions, leading to improved solution quality and scalability. The model achieves strong empirical results, outperforming prior SOTA while using little data and parameters.

**Strengths:**

1.	The paper convincingly argues that MAPF requires modeling joint dependencies beyond pairwise interactions. The use of hypergraphs is theoretically justified as a natural representation of group interactions, which is both elegant and underexplored in this context.

2.	HMAGAT extends the MAGAT architecture by replacing GNN layers with hypergraph attention modules and introducing dynamic hypergraph generation strategies. The directionality and adaptive nature of these hyperedges introduce inductive biases that align well with MAPF’s structure.

3.	The experimental section is robust, spanning multiple map types (small, sparse, dense, and large). HMAGAT demonstrates substantial gains in Sum of Costs (SoC) and success rates, especially in high agent-count and dense environments. The runtime analysis shows HMAGAT is faster and more scalable than MAPF-GPT while maintaining solution quality.

**Weaknesses:**

1.	While the motivation is clear, the paper lacks a formal analysis of why and under what conditions hypergraph attention improves over pairwise attention. A discussion of expressivity (e.g., via Weisfeiler-Lehman hierarchy or permutation invariance properties) could strengthen the theoretical foundation.

2.	Although HMAGAT scales better than MAPF-GPT, it is not evident how hypergraph construction costs scale with agent counts beyond the tested benchmarks. The paper would benefit from a complexity analysis or ablation of hypergraph size versus performance.

3.	All evaluations are performed on Pogema-like benchmarks. It remains unclear how HMAGAT performs on out-of-distribution environments (e.g., dynamic obstacles, non-grid worlds). Including cross-domain tests or transfer experiments would strengthen the empirical story.

4.	The ablation study focuses on performance comparisons but could better disentangle the contributions of hypergraph generation, attention mechanisms, and training data reduction. It’s difficult to isolate which aspect contributes most to the gains.

**Questions:**

1.	How are the hyperedges dynamically generated during training and inference? Are they based purely on spatial proximity, or do they incorporate learned attention or communication priors?

2.	Can the authors formalize or provide intuition for how HMAGAT captures higher-order dependencies that are provably beyond the capacity of pairwise GNNs or transformers?

3.	What is the asymptotic complexity of HMAGAT’s message passing with respect to the number of agents and hyperedges? Is there a practical limit where hypergraph construction becomes prohibitive?

---

> ### Author Response · Authors · 2025-11-21
>
> Dear Reviewer,
>
> Thank you for your insightful comments. Below, we provide our responses. We are still working on additional experiments and analyses, and will provide them later.
>
> **Q1 (hyperedge generation):** We introduce three hypergraph construction strategies: (A) Lloyd, (B) k-means, and (C) shortest-distance-based. These methods rely purely on geometric relationships and do not involve any learning components. Method (C) requires recomputation as agents move. On the other hand, (A) and (B) involve colouring of the grids which is only done once per instance and is independent of the number of agents. After this, the hypergraph generation only requires a quick lookup during runtime. The paper already provides the time complexity analysis; for example, the k-means method requires $O(|k|\times |V|)$.
>
> **Q2 & W1 (formalisation):** We will answer this in the upcoming days.
>
> **Q3 & W2 (aymptotic complexity):** Each agent receives $m$ messages and sends $n$ messages, where $m$ and $n$ are the number of hyperedges with $i$ in the head and tail, respectively (see Fig. 2c). Regarding hypergraph construction cost, as clarified in Q1, in the effort for k-means-based hypergraphs, the colouring depends on the workspace $G$ and is independent of the agent team $A$. With a moderate graph size, this cost is not prohibitive with respect to the model inference. E.g., on the Dense Warehouse map with 128 agents, HMAGAT takes on average 19s to solve, out of which 0.04s are spent on the colouring of the map. 0.005s (~8% of inference time) are spent on constructing the hypergraphs from the colouring at each time step. In practice, this can be highly optimized using C++. We will clarify this in the updated version of the paper.
>
> **W3 (out-of-distribution evaluation):** We focus on standard MAPF due to its well-defined and well-studied nature, as well as its high applicability to real-world applications. POGEMA is simply an interface for MAPF specification. We appreciate the reviewer’s suggestion to extend our work, but it is beyond the scope of this paper.
>
> **W4 (contribution of components):**  We have already presented the effect of these technical components, e.g., hypergraph generation is available in Fig 4. Attention mechanism difference between GNN and HGNN is available in Sec 4.2. The training data is of the same size as MAGAT, i.e. there is no reduction with respect to the base method. Compared to MAPF-GPT, we require a fraction of the data. This is evidence that hypergraphs are an effective representation to capture the complex interactions. Additional data would only help the model learn better, at the cost of being resource intensive.

---

> ### Author Response · Authors · 2025-11-26
> **Official Comment by Authors answering Remaining Questions**
>
> Dear Reviewer,
>
> Thank you for your patience. Below, we provide our responses to the remaining questions.
>
> **W1 (formal analysis of expressive power):** Hypergraphs are a generalization of graphs, where edges (called hyperedges) can connect any number of nodes, rather than just two. It is well-known that hypergraphs are strictly more expressive than graphs, as any graph can be represented as a hypergraph (with all hyperedges connecting exactly two nodes). It is trivial to construct non-isomorphic hypergraphs that will map to the same graph. Thus, from a purely representational standpoint, hypergraphs are trivially more expressive than graphs.
>
> We note that we cannot directly compare the expressive power of GNNs and HGNNs using Weissfeiler-Lehman (WL) tests, as WL tests are designed to compare the expressive power of graph-based models. Works have extended WL tests to hypergraphs, but these extensions are for comparing different hypergraph models, and do not provide a way to compare graph and hypergraph models directly, as they operate on different input spaces.
>
> We also note that both the GNN in MAGAT and the HGNN in HMAGAT are permutation invariant models by design.
>
> Below, we provide some additional discussion from a MAPF perspective.
>
> **Q2 (pairwise interaction’s limitation):** We first note that MAPF is a NP-hard problem, involving complex highly coupled multi-agent dynamics. There does not exist a known ground truth representation for the inter-agent interactions. Graphs (pairwise) and hypergraphs (group) are two ways of modelling these interactions. In Fig. 1(c), we show how a purely pairwise perspective of the interactions leads to sub-optimal solutions compared to a group interaction model. These indicate that hypergraphs should be a better representation for MAPF than graphs, which is further supported by the empirical evidence provided in Sec. 4.
>
> Now, even though MAPF involves complex group interactions, as noted in Section 3, methods using pairwise interactions can still perform well on MAPF tasks, by learning to approximate group interactions. They can do so by: (i) using high-dimensional embeddings that can simply encode all the information about other agents, followed by a complex enough function to decode this information, or (ii) stacking multiple layers of pairwise interactions, which can allow for indirect capture of group interactions. Both MAGAT and MAPF-GPT use a combination of these two strategies. However, both of these strategies have their limitations.
>
> The first strategy relies on the model's ability to encode and decode all relevant information about other agents within high-dimensional embeddings. This may work well for small-scale problems, but as the number of agents increases, the amount of information that needs to be encoded grows significantly, making it increasingly difficult for the model to effectively capture all necessary interactions. The second strategy, stacking multiple layers of pairwise interactions, can help indirectly capture group interactions, but given a fixed number of layers, there is a limit to the complexity of group interactions that can be effectively modeled. As the number of agents increases, the depth required to capture all relevant group interactions may exceed practical limits.
>
> In contrast, hypergraph-based approaches can directly model the group interactions among agents, allowing for a more natural and efficient representation of the MAPF problem. Also, purely from an alignment perspective, hypergraph-based approaches align better with the underlying structure of interactions in optimal MAPF solutions.

---

> > ### Comment · Reviewer_VXsq · 2025-11-27
> >
> > Thank you for your clarification. I keep my score as is.

---

### Official Review · Reviewer_5P7E · 2025-11-01

**Soundness:** 3
**Presentation:** 3
**Contribution:** 2
**Rating:** 4
**Confidence:** 4

**Summary:**

The paper aims to address the limitation of GNN-based approached for MAPF that relies on message passing between pairs of agents. However, in dense environments, it requires higher-order group interaction where a large number of irrelevant neighbors reduce the focus on critical agents. The paper proposes HMAGAT, which is an imitation learning framework that uses HGNN to model group dynamics. The architecture is a CNN-> HGNN -> MLP. The hyper graph generation strategies include k-mean, Lloyd’s alogirhtm and shortest-distance based heuristics. Empirically, HMAGAT outperforms state-of-the-art with smaller network sizes and less training data.

**Strengths:**

1. It is a novel application of hypergraphs to scale on large MAPF instances.
2. The integration of multiple hyper graph generation methods is a new contribution.
3. The analysis of HGNN vs GNN provides a good insight into why HGNN is better than GNN.

**Weaknesses:**

1. The main weakness is the lack of non-ML baselines, such as simple methods like priority planning and more advanced solvers like Large Neighborhood Search or LACAM mentioned in the paper. Those are very efficient and effective solvers that can scale to instances as large as those evaluated in the paper.
2. It isn’t clear how the Rel. SoC metric is computed. See questions.

**Questions:**

When you compute the Rel. Soc, how do you deal with the instances that are unsolved, especially when a. different solvers solve a different set of instances?

**Details Of Ethics Concerns:**

None.

---

> ### Author Response · Authors · 2025-11-21
>
> Dear Reviewer,
>
> Thank you for your insightful comments. Below, we provide our responses.
>
> **W1 (non-learned baseline):** LaCAM3, a state-of-the-art solver in this domain, is already included as a baseline for evaluating solution quality (see Figure 4, caption).  We also remark that a very recent paper (https://arxiv.org/abs/2510.17382) demonstrates that plug-in use of a GNN-based planner within search can outperform established solvers such as LaCAM3. This provides a strong incentive to study better learning architectures for MAPF.
>
> **W2 & Q1 (Rel. SoC):** We follow the strategy used by POGEMA, and, in turn, MAPF-GPT, where the costs for agents that fail to reach their goals are taken to be the episode length. For iterative solvers, this is a valid strategy to get a lower bound on the SoC if the episode length was not limited. LaCAM3 is the only search-based solver evaluated, and it succeeds in all the evaluated scenarios, leading to no issues.

---

### Author Response · Authors · 2025-11-21

Dear Reviewers,

Thank you for your constructive comments. We were pleased to see that the reviewers found our application of hypergraphs to MAPF to be ‘novel’ (5P7E), ‘elegant’ (VXsq) and ‘valuable’ (YpAe), with our model showing ‘strong empirical performance’ (KnuG) and demonstrating ‘substantial gains’ (VXsq). We were also glad that the reviewers found our evaluations to be ‘comprehensive’ (YpAe), ‘diverse’ (KnuG) and ‘robust’ (VXsq). Below, we provide responses to most of your questions so that we can begin discussing them with you. We are still working on several experiments and analyses that we plan on posting in the coming days, alongside our revised paper. Thank you for your patience!

We are grateful to the reviewers for the many constructive insights, and we welcome any further questions or suggestions.

---

### Author Response · Authors · 2025-11-27
**Official Comment by Authors regarding the updated paper**

We thank the reviewers again for their insightful comments. We have updated our paper with the additional results and proofs we have provided in our responses. We have added the latexdiff resulting from our updates in the supplementary materials (diff.pdf). Below, we note the changes we've made in response to the reviewers' comments.

1. **Informal Proof for Attention Dilution:** We have added an informal proof in Appendix E, demonstrating how GNNs suffer from attention dilution, while HGNNs can mitigate this issue.
2. **Clarification on Pairwise Interactions for MAPF:** We have added clarifications on pairwise interaction modelling for MAPF and why they may be insufficient in Appendix F.
3. **Failure Mode Analysis:** We have added a failure mode analysis for HMAGAT in Appendix G.
4. **Further Attention Dilution Experiments:** We have added further experiments on attention dilution, showing that it occurs systematically across different map types and HGNNs can mitigate them, in Appendix H.
5. **Hyperparameter Sensitivity Analysis:** We have added a hyperparameter sensitivity analysis for HMAGAT in Appendix I.
6. **SoC Clarification:** We have clarified how we compute the sum-of-costs (SoC) for failed instances in the evaluation section, Section 4.
7. **Hypergraph Generation Runtime:** We have added the runtime requirement for the hypergraph generation process in Appendix A.2.

We hope these changes address the reviewers' concerns and improve the clarity of our work. We look forward to any further feedback.

---

### Meta-Review · Area_Chair_Hbzt · 2026-01-04

**Summary:**

I will list the most important comments that the reviewers noted during the review process:
1) The lack of non-ML baselines, such as simple methods like priority planning and more advanced solvers like Large Neighborhood Search or LACAM.
2) The paper lacks a formal analysis of why and under what conditions hypergraph attention improves over pairwise attention.
3) It is not evident how hypergraph construction costs scale with agent counts beyond the tested benchmarks.
4) It remains unclear how HMAGAT performs on out-of-distribution environments.
5) It’s difficult to isolate which aspect contributes most to the gains.
6) While proposed combination achieves strong performance, it raises concerns about the overall simplicity and generality of the approach.
7) It is not entirely fair to apply modifications to existing approaches without conducting a proper ablation study.
8) It seems to be an overstatement in the experimental results that HMAGAT consistently outperforms larger models.
9) While the motivation is sound, the execution lacks depth.
9) The failure analysis is missing.
10) The hypergraph generation lacks principled justification.

**Reviewer Concerns:**

The authors did a lot of work during the rebuttal phase and addressed a significant part of the comments:
1) Non-ML baselines: LaCAM3, a state-of-the-art solver in this domain, is already included as a baseline.
2) Asymptotic complexity: the authors give additional explanations.
3) Ablation: authors have presented the effect of technical components, e.g., hypergraph generation is available in Fig 4.
4) PIBT collision shielding: authors gave new data, showing the success rate of MAPF-GPT models w/ and w/o PIBT collision shielding on the highest agent density evaluations.
5) Generality of the approach: the authors mentioned that the mechanism to prepare observations and the temperature sampling are standard and can be easily adapted for imitation learning on other multi-agent tasks.
6) Attention dilution theoretical analysis: the authors provided an informal analysis, demonstrating that GNNs face attention dilution, which HGNNs are capable of mitigating
7) Failure analysis: the authors provided an analysis of the failure cases of HMAGAT on the Dense Room maps.
8) Hypergraph generation: the authors provided three examples (Lloyd, k-means and shortest distance-based) and empirically compared them.

However, some of the comments remained insufficiently addressed:
1) Out-of-distribution performance: the authors have reserved this comment for further research, but it should be an important part of the current work.
2) HMAGAT performance: although the authors corrected themselves in the rebuttal phase regarding the claim of the complete superiority of this model, however, the experimental results do not fully confirm the conclusions of the authors.

The paper is borderline, but after the rebuttal improvements, I'm inclined to accept it.

**Reviewer Scores:**

1) Reviewer 5P7E (score 4) could raise his score.
2) Reviewer VXsq (score 6) explicitly confirmed that he would leave his initial score.
3) Reviewer KnuG (score 4) would most likely have left his initial score.
4) Reviewer YpAe (score 4) could raise his score.

---

### Decision · Program_Chairs · 2026-01-26

Accept (Poster)